# Identifiable Exchangeable Mechanisms for Causal Structure and Representation Learning

**Patrik Reizinger**[*1,3]**, Siyuan Guo**[*1,2]**, Ferenc Huszár**[2]**, Bernhard Schölkopf** [†1,3]**, and Wieland Brendel** [†1,3,4]

[1]Max Planck Institute for Intelligent Systems, Tübingen, Germany
[2]University of Cambridge, Cambridge, United Kingdom
[3]ELLIS Institute Tübingen, Tübingen, Germany
[4]Tübingen AI Center, Tübingen, Germany

## Abstract

Identifying latent representations or causal structures is important for good generalization and downstream task performance. However, both fields developed rather independently. We observe that several structure and representation identifiability methods, particularly those that require multiple environments, rely on exchangeable non–i.i.d. (independent and identically distributed) data. To formalize this connection, we propose the Identifiable Exchangeable Mechanisms (IEM) framework to unify key representation and causal structure learning methods. IEM provides a unified probabilistic graphical model encompassing causal discovery, Independent Component Analysis, and Causal Representation Learning. With the help of the IEM model, we generalize the Causal de Finetti theorem of Guo et al. (2024a) by relaxing the necessary conditions for causal structure identification in exchangeable data. We term these conditions cause and mechanism variability, and show how they imply a duality condition in identifiable representation learning, leading to new identifiability results.

## 1 Introduction

Provably identifying latent representations and causal structures has been a central problem in machine learning, as such guarantees promise good generalization and downstream task performance (Richens & Everitt, 2024; Perry et al., 2022; Zimmermann et al., 2021; Arjovsky et al., 2020; Arjovsky, 2021; Brady et al., 2023; Wiedemer et al., 2023b;a; Lachapelle et al., 2023; Rusak et al., 2024). Causal structure identification, also known as Causal Discovery (CD), aims to infer cause-effect relationships, whereas identifiable representation learning aims to infer ground-truth sources. Due to their different learning objectives, such problems have been treated separately.

Recent works on Causal Representation Learning (CRL) (Schölkopf et al., 2021) propose to learn latent representations with causal structures that allow efficient generalization in downstream tasks. Yet despite progress (Zečević et al., 2021; Reizinger et al., 2023; Xi & Bloem-Reddy, 2023), our understanding is still limited regarding the question of

*what enables structure and representation identifiability?*

Guo et al. (2024a) formalize causality for exchangeable data generating processes (DGPs), showing that unique structure identification is feasible under exchangeable non–i.i.d. data, assuming Independent Causal Mechanisms (ICMs) (Schölkopf et al., 2012). Such unique structure identification was classically deemed impossible (Pearl, 2009a). The present work makes the observation that exchangeable non–i.i.d. data is the driving force in identification for both representation and structure identification. We introduce a unified framework for CD, Independent Component Analysis (ICA), and CRL (Fig. 1) and show that relaxed exchangeability conditions, termed cause and mechanism variability (Fig. 2), are sufficient for both representation and structure identifiability. Our contributions are:

- **Unifying structure and representation learning under the lens of exchangeability (§ 3, also cf. Fig. 1)**: We develop a probabilistic model, Identifiable Exchangeable Mechanisms (IEM), that subsumes key methods in CD, ICA, and CRL.
- **Relaxing causal discovery assumptions in exchangeable non–i.i.d. data (§ 3.2)**: we show how exchangeable non–i.i.d. cause or effect-given-cause mechanisms, termed *cause and mech-*

---

*Equal contribution. Correspondence to `patrik.reizinger@tuebingen.mpg.de`
†Equal supervision

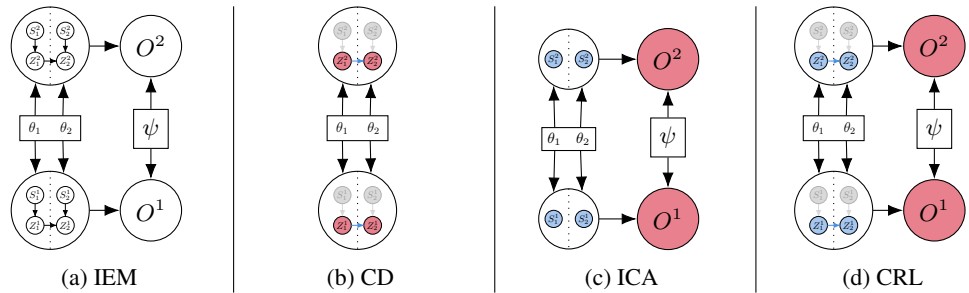

(a) IEM      (b) CD      (c) ICA      (d) CRL

Figure 1: **Identifiable Exchangeable Mechanisms (IEM)–A unified model for structure and representation identifiability:** Here we show that exchangeable but non-i.i.d. data enables identification in key methods across Causal Discovery (CD), Independent Component Analysis (ICA), and Causal Representation Learning (CRL). Fig. 1a shows the graphical model for IEM, which subsumes Causal Discovery (CD) (§ 3.2), Independent Component Analysis (ICA) (§ 3.3), and Causal Representation Learning (CRL) (§ 3.4). $S$ denotes latent, $Z$ causal, and $O$ observed variables with corresponding latent parameters $\boldsymbol{\theta}, \psi$, superscripts denote different samples. Red denotes observed/known quantities, blue stands for target quantities, and gray illustrates components that are *not* explicitly modeled in a particular paradigm. $\theta_i$ are latent variables controlling separate probabilistic mechanisms, indicated by dotted vertical lines. **CD** (Fig. 1b) corresponds to the left-most layer of IEM, focusing on the study of cause-effect relationships between observed causal variables; **ICA** (Fig. 1c) infers source variables from observations, but without causal connections in the left-most layer of IEM; **CRL** (Fig. 1d) shares the most similar structure with IEM, as it has both layers, including the intermediate causal representations. See Fig. 4 for an enlarged view

*anism variability*, provide sufficient and necessary conditions for bivariate CD, generalizing the identification theorem in (Guo et al., 2024a).

- **Providing dual identifiability results in Time-Contrastive Learning (TCL) (§ 3.3)**: we show how an auxiliary-variable ICA method, TCL, is a special case of cause variability—we discuss Generalized Contrastive Learning (GCL) in Appx. A.3. Using insights from the duality in cause and mechanism variability, we prove the identifiability of TCL under mechanism variability.

## 2   PRELIMINARIES

The impossibility of bivariate CD (Pearl, 2009b) and representation identifiability (Hyvärinen & Pajunen, 1999; Locatello et al., 2019) from i.i.d. data is well known (cf. Appx. C for examples). Thus, we focus on non–i.i.d., particularly, exchangeable data (Defn. 1) and discuss a causal framework from (Guo et al., 2024a) building on exchangeability. An example of exchangeable non–i.i.d. data is when training samples come from different distributions, e.g., Gaussians with different means and/or variances, where the different means and/or variances are modeled as (causal) de Finetti parameters.

**Notation.** Capital letters denote random variables (RVs), lowercase letters their realizations, and bold letters sets/vectors of RVs. $\mathbf{S}$ are the latent sources in representation learning or, equivalently, the set of exogenous variables in a Structural Equation Model (SEM); $\mathbf{Z}$ are causal variables, and $\mathbf{O}$ are observations in (causal) representation learning. Data generated by a DGP is a sequence of RVs $\mathbf{X}^1, \mathbf{X}^2, \ldots$ where superscripts index samples and subscripts the vector components (RVs), i.e., $X_i^1$ specifies the $i^{th}$ random variable in $\mathbf{X}^1$. $f$ is the mixing function between latents to observations, i.e., $f : \mathbf{s} \to \mathbf{o}$ for representation learning, and $f : \mathbf{z} \to \mathbf{o}$ for CRL. Structural assignments from exogenous to causal variables are denoted as $Z := g(\mathbf{Pa}(Z))$, where $\mathbf{Pa}(Z)$ are the parents or causes of $Z$ and $\mathbf{Pa}(Z)$ includes the corresponding exogenous variable $S$. Whenever the RV sequence contains a single variable or bivariate pairs per position, we use $X^n$ or $(X^n, Y^n)$. Uppercase $P$ is a probability distribution, and lowercase $p$ is a probability density function. $\delta_{\theta_0}(\theta)$ is a shorthand for the delta-distribution $\delta(\theta = \theta_0)$.

**Causal de Finetti (CdF) and Exchangeability.**

**Definition 1** (Exchangeable sequence). *An infinite sequence of random variables $X^1, X^2, \ldots$ is exchangeable if for any finite permutation $\pi$ on the position indices, the joint distribution satisfies:*

$$P(X^1, \ldots, X^n) = P(X^{\pi(1)}, \ldots, X^{\pi(n)}). \tag{1}$$

An important result to characterize any exchangeable sequence is the theorem of de Finetti (1931). It states that for any exchangeable sequence, there exists a latent variable $\theta$ such that the sequence's joint distribution can be represented as a mixture of *conditionally* i.i.d. distributions:

$$P(x^1, \ldots, x^n) = \int \prod_{i=1}^{n} p(x^i|\theta)p(\theta)d\theta. \qquad (2)$$

Any i.i.d. sequence is exchangeable since $p(x^1, \ldots, x^n) = \prod_{i=1}^{n} p(x^i)$ and the joint distribution remains identical when changing the order of observations. Alternatively, the right-hand side of Eq. (2) collapses to an i.i.d. sequence whenever $p(\theta) = \delta(\theta = \theta_0)$ for some constant $\theta_0$. Though i.i.d. is a special case of exchangeable sequences, not all exchangeable sequences are i.i.d. Examples include, but are not limited to: the Pólya urn model (Hoppe, 1984), Chinese restaurant processes (Aldous et al., 1985), or Dirichlet processes (Ferguson, 1973).

**Causality.** Causality infers the ground-truth causal structure from the observed joint distribution $P$ to enable efficient generalization in novel scenarios. It studies interventional and counterfactual queries beyond purely associational relationships in observational data. The ICM principle (Schölkopf et al., 2021) hypothesizes that distinct causal mechanisms neither inform nor influence each other. Guo et al. (2024a) proves that for exchangeable sequences, the ICM principle implies the existence of statistically independent latent variables governing each causal mechanism. Thus, establishing a mathematical framework to study causality in exchangeable data. We state their bivariate result.

**Theorem 1** (Causal de Finetti (Guo et al., 2024a)). *Let $\{(X^n, Y^n)\}_{n \in \mathbb{N}}$ be an infinite sequence of binary random variable pairs and denote the set $\{1, 2, \ldots, n\}$ as $[n]$. The sequence is infinitely exchangeable, and satisfies $Y^{[n]} \perp X^{n+1} \mid X^{[n]}$ for all $n \in \mathbb{N}$ if and only if there exists random variables $\theta \in [0, 1]$ and $\psi \in [0, 1]^2$ such that the joint probability can be represented as*

$$P(x^1, y^1, \ldots, x^n, y^n) = \int \prod_{i=1}^{n} p(y^i \mid x^i, \psi)p(x^i \mid \theta)p(\theta)p(\psi)d\theta d\psi \qquad (3)$$

Thm. 1 shows that for any exchangeable sequence with paired random variables that satisfy certain conditional independences, there exist statistically independent latent variables $\theta, \psi$ governing each (causal) mechanism $P(Y \mid X, \psi), P(X|\theta)$. Guo et al. (2024a) further shows that unique causal structure identification is possible in exchangeable non–i.i.d. data, contrary to the common belief that structure identification is infeasible in i.i.d. data (Pearl, 2009a).

Our work further observes that exchangeable non–i.i.d. data is again the key for representation identifiability. We thus propose our unifying model, IEM, to allow understanding the driving forces behind general identifiability.

## 3 IDENTIFIABLE EXCHANGEABLE MECHANISMS: A UNIFYING FRAMEWORK FOR STRUCTURAL AND REPRESENTATIONAL IDENTIFIABILITY

This section demonstrates how non–i.i.d., particularly, exchangeable data (Defn. 1) enables several structure and representation identifiability results. We introduce Identifiable Exchangeable Mechanisms (IEM) (cf. Fig. 1 and § 3.1) to illustrate how exchangeability is the common principle for multiple identifiability results across Causal Discovery (CD), Independent Component Analysis (ICA), and Causal Representation Learning (CRL). Furthermore, we relax the exchangeability condition into what we call *cause and mechanism variability,* which provides novel and relaxed identifiability conditions (Thm. 2 and Lem. 4). We derive a probabilistic model from IEM for CD, ICA, and CRL (see the graphical relationship in Fig. 1). Then, we show how the bivariate Causal de Finetti (CdF) theorem (Guo et al., 2024a) (§ 3.2), TCL (Hyvarinen & Morioka, 2016) (§ 3.3), and CauCA (Wendong et al., 2023) (§ 3.4) all leverage exchangeable data, and, thus, are special cases of IEM.

### 3.1 IDENTIFIABLE EXCHANGEABLE MECHANISMS (IEM)

IEM encompasses three types of variables: exogenous (source) variables $\mathbf{S}$ for (disentangled) latent representations, causal variables $\mathbf{Z}$ for representations that contain cause–effect relationships, and observed variables $\mathbf{O}$ for observed (high-dimensional) quantities.

**A probabilistic model for IEM.** With all three variable types, assuming that there is an intermediate causal layer, the joint distribution of source, causal, and observed variables is:

$$p(\mathbf{s}, \mathbf{z}, \mathbf{o}) = \int_{\boldsymbol{\theta}^s, \boldsymbol{\theta}^g, \boldsymbol{\psi}} p(\mathbf{o}|\mathbf{z}, \boldsymbol{\psi}) \prod_j [p(z_j|\mathbf{Pa}(z_j); \theta_j^g)p(\theta_j^g)] \prod_i [p(s_i|\theta_i^s)p(\theta_i^s)] \, p(\boldsymbol{\psi}) d\boldsymbol{\psi} d\boldsymbol{\theta}^g d\boldsymbol{\theta}^s, \quad (4)$$

where $j$ indexes causal, $i$ source variables (we omit the sample superscript for brevity), $\mathbf{Pa}(z_j)$ denotes the parents of $z_j$ (including $s_j$) and we integrate over all $\theta_j^g$ and $\theta_i^s$—the superscripts $g$ and $s$ denote separate parameters controlling structural assignments $g_j$ and the source distributions, respectively.

**An intuition for IEM.** Consider multi-environment data where each environment has a distinct distribution, while observations within the same environment are assumed to be exchangeable, i.e., the observations' order is irrelevant. IEM models such multi-environment data by treating each environment as an i.i.d. copy of the model in (4). Across-environment variability is ensured by choosing non-delta parameter priors $p(\boldsymbol{\psi}), p(\theta_j^g), p(\theta_i^s)$, while exchangeability within the environment is ensured by the conditional independence of observations given these parameters. i.i.d. data, or a single environment in this context, is a special case of exchangeable data with delta priors (see § 2).

*We introduce IEM to elucidate the relationship of CD, ICA, and CRL: despite distinct learning objectives, they often rely on the same exchangeable non–i.i.d. data structure to allow structure or representation identification.*

Further, IEM can model both the passive (observation) and active (intervention) view of data. For example, both a passive distribution shift and an active hard intervention can be modelled with exchangeability as a switch between binary variables. Tab. 1 in Appx. D illustrates the similarity of the (passive) variability and (active) interventional assumptions.

The graphical model of IEM illustrates the relationship of source, causal and observed variables (Fig. 1). We connect the seemingly unrelated methods of CD, ICA, and CRL by deriving their model from IEM via *omission* (cf. Figs. 1b to 1d). Namely, CD does not handle high-dimensional observations, ICA does not model causal variables, and CRL does not aim to recover source variables. We detail these connections in the following case studies.

**Case study: Identifiable Latent Neural Causal Models (Liu et al., 2024) in the unified model.**
Liu et al. (2024) proposed to learn source (exogenous) variables, causal variables, and the corresponding Directed Acyclic Graph (DAG) together, i.e., all target quantities from Fig. 1. We show how this is possible via exchangeable sources and mechanisms (Lem. 1). For the sources, they assume non-stationary, conditionally exponential source variables. Thus, they can use TCL (Hyvarinen & Morioka, 2016) to identify $\mathbf{S}$ from $\mathbf{O}$ (details in § 3.3). For the causal variables, they require diverse interventions, quantified by a derivative-based condition (Assum. 1) on the structural assignments $g_i$ (the authors generalize to post-nonlinear models; we focus on Additive Noise Models (ANMs)).

**Assumption 1** (Structural assignment assumption (Liu et al., 2024)). *Assume that the structural assignments $g_i$ between causal variables $z_i$ form an ANM such that $z_i := g_i(Pa(z_i); \theta_i^g(u)) + s_i$, where $\theta_i^g(u)$ are the parameters of the structural assignments, and they depend on the auxiliary-variable $u$. Then, to identify the causal structure and causal variables, there exists a value $u = u_0$ such that (denoting $\theta_{i0}^g := \theta_i^g(u_0)$)*

$$\forall z_j \in \boldsymbol{Pa}(z_i): \quad \frac{\partial g_i(\boldsymbol{Pa}(z_i), \theta_i^g = \theta_{i0}^g)}{\partial z_j} = 0. \tag{5}$$

Assum. 1 requires for a specific value $u = u_0$, the path $Z_j \to Z_i$ for each $Z_j \in \mathbf{Pa}(Z_i)$ is blocked— this can be thought of as emulating perfect interventions, for which structure identifiability results exists (Pearl, 2009b). We rephrase the identifiability result of Liu et al. (2024), showing how it relies on exchangeability conditions (see Appx. A.7 for proof):

**Lemma 1.** *[Identifiable Latent Neural Causal Models are identifiable with exchangeable sources and mechanisms] The model of Liu et al. (2024) (Fig. 1a) identifies both the latent sources $\mathbf{s}$ and the causal variables $\mathbf{z}$ (including the graph), by the variability of $\mathbf{s}$ via a non-delta prior over $\theta^s$ and by the variability of the structural assignments via $\theta^g$.*

The identifiability result of (Liu et al., 2024) requires two separate variability conditions: one for the sources and one for the mechanisms. We show how these separate conditions, when the SEM is an ANM, disentangle the CdF parameters into separate (independent) parameters controlling sources and structural assignments respectively (see proof in Appx. A.8):

**Lemma 2.** *[Independent source and structural assignment CdF parameters for ANMs] In the setting of Liu et al. (2024), where the SEM is an ANM, the CdF parameters for the sources, $\boldsymbol{\theta^s}$, and the structural assignments, $\boldsymbol{\theta^g}$, are independent, i.e. $p(\boldsymbol{\theta^g}, \boldsymbol{\theta^s}) = p(\boldsymbol{\theta^g})p(\boldsymbol{\theta^s})$.*

Lem. 2 says that the representation learning (TCL) part relies on the exchangeability of the source (exogenous) variables, whereas the CRL part requires exchangeability in the SEM. The connection between Gaussian LTI systems and CdF (Rajendran et al., 2023, Sec. 3.5) can be seen as a special

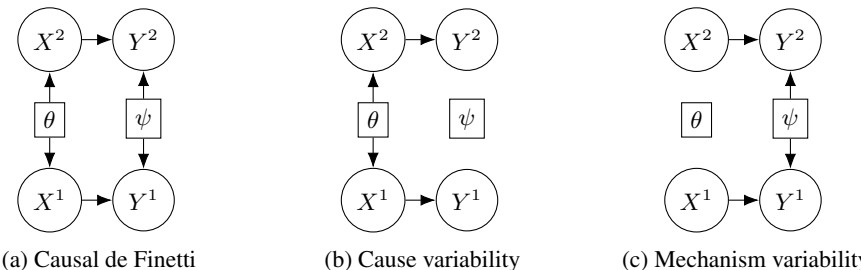

|  (a) Causal de Finetti | (b) Cause variability | (c) Mechanism variability |

Figure 2: **non–i.i.d. conditions for bivariate CD: (a)** Exchangeable non–i.i.d. DGP for both cause $P(\mathbf{X})$ and mechanism $P(\mathbf{Y}|\mathbf{X})$ (Guo et al., 2024a); **(b):** exchangeable non–i.i.d. DGP for cause $P(\mathbf{X})$ and i.i.d. DGP for mechanism $P(\mathbf{Y}|\mathbf{X})$ **(c):** exchangeable non–i.i.d. DGP for mechanism $P(\mathbf{Y}|\mathbf{X})$ and i.i.d. DGP for cause $P(\mathbf{X})$. Thm. 2 shows that identifying the unique bivariate causal structure is possible if either the cause or the mechanism follows an exchangeable non–i.i.d. DGP

case of Lem. 2, where the sources and the mechanism (the LTI dynamical system) have independent matrix parameters. Our result also conceptually resemble mechanized SEMs (Kenton et al., 2023), where the structural assignments are modeled by distinct nodes.

Next, we show how the probabilistic models for CD, ICA, and CRL can be derived from IEM (Fig. 1), depending on whether we model cause-effect relationships and/or source variables.

## 3.2 EXCHANGEABILITY IN CAUSAL DISCOVERY: EXTENDING CAUSAL DE FINETTI

Causal Discovery (CD) infers the causal graph between observed *causal* variables (Fig. 1b). SEMs (Pearl, 2009a) are classic causal models, where deterministic causal mechanisms and stochastic noise (exogenous/latent) variables determine each causal variable's value. For i.i.d. observational data alone, causal structure is identifiable only up to its Markov equivalence class (Defn. 5). In the present work, we introduce a relaxed set of conditions, termed cause and mechanism variability, and show in the bivariate case how these non–i.i.d., specifically a mixture of i.i.d. and exchangeable, data, are necessary and sufficient for uniquely identifying causal structures.

**A probabilistic model for CD.** We consider the bivariate case with exchangeable sequences $(X^n, Y^n)$ that adheres to the ICM principle (Peters et al., 2018). Thm. 1 states there exist statistically independent CdF parameters $\theta, \psi$ such that the joint distribution can be represented as:

$$p(x^1, y^1, \ldots, x^n, y^n) = \int_\theta \int_\psi \prod_{i=1}^n p(y^i|x^i, \psi)p(x^i|\theta)d\psi d\theta \quad \text{where} \quad \psi \perp \theta. \qquad (6)$$

CD with unique structure identification is possible when the parameter priors $p(\theta), p(\psi)$ are not delta distributions, i.e., when the data pairs are from exchangeable non–i.i.d. sequences. Fig. 2a shows a Markov graph compatible with (6).

**Case study: CdF in the unified model.** CD in general, and CdF in particular, focuses on the study of *observed* causal variables (denoted by $Z$ in Fig. 1 and (4)). CD aims to learn cause-effect relationships among the observed causal variables $Z_i$, rather than reconstructing the $Z_i$ or uncovering the true mixing function $f$. Bivariate CdF fits into the IEM probabilistic model by relabeling $Y = Z_i$ and $X = \mathbf{Pa}(Z_i)$. We use our insights from IEM to relax the assumptions for bivariate CD from exchangeable pairs, generalizing CdF.

**Relaxing CdF: cause and mechanism variability.** We show that it is not necessary for CD that *both* $p(\theta)$ and $p(\psi)$ differ from a delta distribution—equivalently, the presence of both graphical substructures $X^1 \leftarrow \theta \rightarrow X^2$ and $Y^1 \leftarrow \psi \rightarrow Y^2$ are *not* required to distinguish the causal direction between $X$ and $Y$. We distinguish two cases: *"cause variability,"* when only the cause mechanism changes (Fig. 2b), i.e. $p(\psi) = \delta_{\psi_0}(\psi), p(\theta) \neq \delta_{\theta_0}(\theta)$; and *"mechanism variability,"* when only the effect-given-the-cause mechanism changes (Fig. 2c), i.e. $p(\theta) = \delta_{\theta_0}(\theta)$ and $p(\psi) \neq \delta_{\psi_0}(\psi)$—we motivate these assumptions by the Sparse Mechanism Shift (SMS) hypothesis (Perry et al., 2022) and provide real-world examples for both in Appx. E. When $p(\theta)$ is sufficiently different from a delta distribution, then each cause distribution sampled from $p(x|\theta)$ will have a different distribution with high probability. This is similar for $p(\psi)$ when the effect-given-the-cause mechanism $p(y|x, \psi)$ is shifted. Formally (the proof is in Appx. A.1):

**Theorem 2.** *[Cause/mechanism variability is necessary and sufficient for bivariate CD] Given a sequence of bivariate pairs $\{X^n, Y^n\}_{n \in \mathbb{N}}$ such that for any $N \in \mathbb{N}$, the joint distribution can be represented as:*

- *$X \to Y$: $p(x^1, y^1, \ldots, x^N, y^N) = \int_\theta \int_\psi \prod_n p(y^n|x^n, \psi) p(x^n|\theta) p(\theta) p(\psi) d\theta d\psi$*
- *$X \leftarrow Y$: $p(x^1, y^1, \ldots, x^N, y^N) = \int_\theta \int_\psi \prod_n p(x^n|y^n, \theta) p(y^n|\psi) p(\psi) p(\theta) d\theta d\psi$*

*Then the causal direction between variables $X, Y$ can still be distinguished when:*

1. *either only $p(\theta) = \delta_{\theta_0}(\theta)$ for some constant $\theta_0$ or only $p(\psi) = \delta_{\psi_0}(\psi)$ for some constant $\psi_0$ (but not both). Fig. 2b and Fig. 2c show the Markov structure of such factorizations.*
2. *the distribution of $P$ is faithful (Defn. 4) w.r.t. Fig. 2b or Fig. 2c.*

Thm. 2 relaxes Thm. 1 and states that the causal structure can be identified even if only one mechanisms varies. That is, if the $X, Y$ pairs are a mixture of i.i.d. and exchangeable data such that either cause variability (Fig. 2b) or mechanism variability (Fig. 2c) holds; then we can distinguish $X \to Y$ from $X \leftarrow Y$—which we empirically verify in synthetic experiments in Appx. F. Thm. 2 focuses on the bivariate case, though we expect similar results can be extended to multivariate cases. Thm. 2 aligns with well-known results stating that assuming no confounders, single-node interventions are sufficient to identify the causal structure (Pearl, 2009b). The contribution of Thm. 2 lies in taking the passive view, similar to (Guo et al., 2024a).

### 3.3 EXCHANGEABILITY IN REPRESENTATION LEARNING

Representation learning aims to infer latent sources $\mathbf{s}$ from observations $\mathbf{o}$, which are generated via a mixing function $f : \mathbf{s} \to \mathbf{o}$. ICA[1] (Comon, 1994; Hyvarinen et al., 2001; Shimizu et al., 2006) assumes component-wise independent latent sources $\mathbf{s}$ where $p(\mathbf{s}) = \prod_i p_i(s_i)$ and aims to learn an unmixing function that maps to independent components. Recent methods (Hyvarinen & Morioka, 2016; Hyvarinen et al., 2019; Khemakhem et al., 2020a; Morioka et al., 2021; Zimmermann et al., 2021) focus on auxiliary-variable ICA, which assumes the existence and observation[2] of an auxiliary variable $u$ such that $p(\mathbf{s}|u) = \prod_i p_i(s_i|u)$ holds—thus, providing identifiability results for a much broader model class. Here, we show that representation identifiability in (auxiliary-variable) ICA, particularly TCL (Hyvarinen & Morioka, 2016) (and GCL with conditionally exponential-family sources, cf. Appx. A.3), relies on the latent sources to be an exchangeable non–i.i.d. sequence.

**A probabilistic model for (auxiliary-variable) ICA.** Auxiliary variables can represent many forms of additional information (Hyvarinen et al., 2019). Our focus is when $u$ represents segment indices, i.e., it enumerates multiple environments. This is equivalent to a draw from a categorical prior $p(u)$, thus, sources are a marginal copy of an exchangeable sequence $p(\mathbf{s}) = \int_u \prod_i p(s_i|u) p(u) du$. In auxiliary-variable ICA (Hyvarinen & Morioka, 2016; Hyvarinen et al., 2019; Khemakhem et al., 2020a), there is a separate parameter $\theta_i := \theta(u)$ for each $s_i$. Conditioned on observing the segment index $u$, the joint probability distribution w.r.t. latent sources $\mathbf{s}$ and observations $\mathbf{o}$ factorizes as ($i$ indexes source variables, we omit the sample superscript for brevity):

$$p_u(\mathbf{o}, \mathbf{s}) = \int_{\boldsymbol{\theta}} \int_{\boldsymbol{\psi}} p(\mathbf{o}|\mathbf{s}, \boldsymbol{\psi}) \prod_i [p(s_i|\theta_i) p_u(\theta_i)] \, d\boldsymbol{\psi} d\boldsymbol{\theta} \quad \text{where} \quad p(\boldsymbol{\psi}) = \delta_{\boldsymbol{\psi}_0}(\boldsymbol{\psi}). \tag{7}$$

Compared to (4), Eq. (7) does not have a "causal layer", expressing the (conditional) independence between the sources in ICA. Compared to CdF, representation learning with ICA additionally restricts the joint probability between sources and observations to extract more information (the latent variables), compared to only the DAG. This relation was demonstrated by Reizinger et al. (2023), showing that representation identifiability in some cases implies causal structure identification.

**Case study: TCL in the unified model.** We next present how auxiliary-variable ICA, particularly TCL (Hyvarinen & Morioka, 2016) (cf. Appx. A.3 for the generalization), fits into IEM (Fig. 1c) and present a duality result on cause and mechanism variability. The TCL model assumes that the conditional log-density $\log p(\mathbf{s}|u)$ is a sum of components $q_i(s_i, u)$, where $q_i$ belongs to the exponential family of order one, i.e.:

$$q_i(s_i, u) = \tilde{q}_i(s_i) \theta_i(u) - \log N_i(u) + \log Q_i(s_i), \tag{8}$$

---

[1]Though the literature is referred to as the nonlinear ICA literature, it often uses *conditionally independent latents*, but expressions such as Independently Modulated Component Analysis (IMCA) are not widely used

[2]There is a variant of auxiliary-variable ICA for Hidden Markov Models, which does not require observing $u$ (Morioka et al., 2021); we focus on the case when $u$ is observed

where $N_i$ is the normalizing constant, $Q_i$ the base measure, $\tilde{q}_i$ the sufficient statistics, and the modulation parameters $\theta_i := \theta_i(u)$ depend on $u$. The identifiability of TCL requires multiple segments (i.e., realizations of $u$ with different values) such that for environment $j$, the modulation parameters fulfill a sufficient variability condition, defined via a rank condition:

**Assumption 2** (Sufficient variability). *A DGP is called sufficiently variable if there exists $(d+1)$ distinct realizations of $u$ for $d-$dimensional source variables and modulation parameter vectors such that the modulation parameter matrix $\mathbf{L} \in \mathbb{R}^{(E-1) \times d}$ has full column rank. For $E$ environments and modulation parameter vectors $\boldsymbol{\theta}^j = \left[\theta_1^j, \ldots, \theta_d^j\right]$, the $j^{th}$ row of $\mathbf{L}$ is:*

$$[\mathbf{L}]_{j:} = (\boldsymbol{\theta}^j - \boldsymbol{\theta}^0). \tag{9}$$

Here $\theta_i$ are the de Finetti parameters for the exchangeable sources. We show in Appx. A.2 that $p_u(\theta_i)$ cannot be a delta distribution; otherwise, the variability condition of TCL is violated. Thus, the identifiability of TCL hinges on exchangeable non–i.i.d. sources (we prove the same for conditionally-exponential sources in GCL (Hyvarinen et al., 2019), cf. Cor. 1 in Appx. A.3):

**Lemma 3.** *[TCL is identifiable due to exchangeable non–i.i.d. sources] The sufficient variability condition in TCL corresponds to cause variability, i.e., exchangeable non–i.i.d. source variables with a fixed mixing function, which leads to the identifiability of the latent sources.*

**Extending TCL via the cause–mechanism variability duality.** We next demonstrate the flexibility of the IEM framework as it relates the probabilistic model for TCL to that of bivariate CdF (6). Treating the observations $\mathbf{o}$ as the "effect", and the source vector $\mathbf{s} = [s_1, \ldots, s_d]$ as the "cause", (7) becomes equal to (6) when $\mathbf{X} = \mathbf{S}$ and $\mathbf{Y} = \mathbf{O}$. As in auxiliary-variable ICA the mixing function $f$ is *deterministic*, it constitutes "cause variability" (Fig. 2b). Our extension of the CdF theorem in Thm. 2 shows a symmetry between cause and mechanism variability: flipping the arrows and relabeling $X/Y$ and $\theta/\psi$ transforms one case into the other (cf. Fig. 5 in Appx. A.1). Our insight is that identification can be achieved both with cause variability or mechanism variability. This not only holds for CD, leading to a dual formulation of TCL with mechanism variability. We illustrate this in an example, then state our result (cf. Appx. A.4 for the proof):

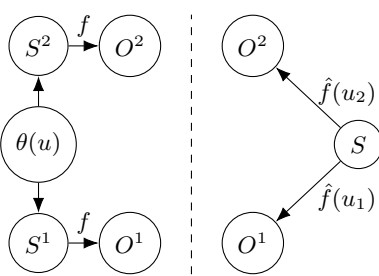

Figure 3: **The duality of cause and mechanism variability in TCL:** Lem. 4 shows that the same identifiability result holds in **(Left):** the original TCL setting with exchangeable non–i.i.d. sources $S$ with deterministic $f$ mixing (cause variability), and the matching **(Right):** i.i.d. sources $S$ with a stochastic $\hat{f}(u)$ mixing (mechanism variability)

**Example 1** (Duality of cause and mechanism variability for Gaussian models). *Assume conditionally independent latent sources with variance-modulated Gaussian components, i.e., $p(\mathbf{s}|u) = \prod_i p_i(s_i|u)$, where each $p_i(s_i|u) = \mathcal{N}\left(\mu_i; \sigma_i^2(u)\right)$, depending on auxiliary variable $u$. In this case, the observation distribution is the pushforward of $p(s|u)$ by $f$, denoted as $f_* p(s|u)$. For given $\sigma_i^2(u)$ and $f$, where $\Sigma^2(u) = \text{diag}\left(\sigma_1^2(u), \ldots, \sigma_n^2(u)\right)$, we can find stochastic functions $\hat{f} = f \circ \Sigma(u)$ such that the pushforward $f_* p(s|u) = f_* \mathcal{N}\left(\boldsymbol{\mu}; \Sigma^2(u)\right)$ equals to $\hat{f}_* \mathcal{N}\left(\boldsymbol{\mu}; \mathbf{I}\right)$. By construction, $\hat{f}$ varies with $u$ and satisfies mechanism variability.*

**Lemma 4.** *[Duality of cause and mechanism variability for TCL] For a given deterministic mixing function $f : \mathbf{s} \to \mathbf{o}$ and conditionally factorizing (non-stationary) latent sources $p(\mathbf{s}|u) = \prod_i p_i(s_i|u)$ fulfilling the sufficient variability of TCL, there exists an equivalent setup with stationary (i.i.d.) sources $p(\mathbf{s}) = \prod_i p_i(s_i)$ with stochastic functions $\hat{f} = f \circ g : \mathbf{s} \to \mathbf{o}$, where $g = g(u)$ and each component $g_i$ is defined as an element-wise function such that the pushforward of $p_i(s_i)$ by $g_i$ equals $p_i(s_i|u)$, i.e., $g_{i*} p_i(s_i) = p_i(s_i|u)$. Then, $g_{i*} p_i(s_i)$ fulfils the same variability condition; thus, the same identifiability result applies.*

Lem. 4 shows that both cause and mechanism variability lead to representation identification in TCL, visualized in Fig. 3. We illustrate the practical differences between cause and mechanism variability in the medical example of learning representations from fMRI data (Hyvarinen & Morioka, 2016; Khemakhem et al., 2020a). Cause variability means having access to data from patients with different

underlying conditions; mechanism variability corresponds to measuring a single patient's condition with multiple diagnostic methods.

## 3.4 EXCHANGEABILITY IN CAUSAL REPRESENTATION LEARNING

Causal Representation Learning (CRL) aims to learn the causal representations $\mathbf{Z}$ and their graphical structure from high-dimensional observations $\mathbf{O}$. That is, CRL can be considered as performing representation learning for the latent causal variables and CD between those learned latent variables simultaneously.

**A probabilistic model for CRL.** Auxiliary-variable ICA considers the source distribution as $\prod_i p(s_i|\theta_i)$, where $s_i$ and $s_j$ are not causally related. CRL takes one step further and studies how to find causal dependencies between the latent causal variables. We show that CdF theorems apply just as de Finetti applies in *exchangeable ICA*. In CRL the joint distribution factorizes ($j$ indexes causal variables, we omit the sample superscript for brevity):

$$p(\mathbf{z}, \mathbf{o}) = \int_{\boldsymbol{\theta}} \int_{\psi} p(\mathbf{o}|\mathbf{z}, \psi) \prod_j [p(z_j|\mathbf{Pa}(z_j); \theta_j)p(\theta_j)] \, d\psi d\boldsymbol{\theta}, \tag{10}$$

where $\theta_j$ are the CdF parameters controlling each latent causal mechanism, leading to exchangeable causal variables that adhere to the ICM principle. Compared to exchangeable ICA (7), eq. (10) allows that the modeled latent causal variables $z_i$ can depend on each other, whereas ICA does not model cause–effect relationships (Fig. 1d).

**Case study: CauCA in the unified model.** Causal Component Analysis (CauCA) (Wendong et al., 2023) defines a subproblem of CRL by assuming that the DAG between the $z_i$ is known. Wendong et al. (2023) show that identifying causal representations $z_i$ requires single-node (soft) interventions that change the probabilistic mechanisms $p(z_j|\mathbf{Pa}(z_j))$ almost everywhere, which they quantify with the interventional discrepancy:

**Assumption 3** (Interventional discrepancy condition (Wendong et al., 2023)). *Pairs of observational and single-node (soft) interventional densities $p, \tilde{p}$ need to differ almost everywhere, i.e.:*

$$\frac{\partial}{\partial z_j} \log \frac{\tilde{p}(z_j|\boldsymbol{Pa}(z_j))}{p(z_j|\boldsymbol{Pa}(z_j))} \neq 0 \quad a.e. \tag{11}$$

Satisfying Assum. 3 means an intervention on the parameters $\theta_j$ of the causal mechanisms $p(z_j|\mathbf{Pa}(z_j); \theta_j)$ (we compare to Assum. 1 in Appx. A.7). The following lemma follows from having interventions on the value of $\theta_j$ that fulfill Assum. 3 (proof in Appx. A.5):

**Lemma 5.** *[Non-delta priors in the causal mechanisms can enable identifiable CRL]*
*If the interventional discrepancy condition Assum. 3 holds, then the parameter priors in (10) cannot equal a delta distribution, i.e., $p(\theta_j) \neq \delta_{\theta_{j0}}(\theta_j)$; thus, if the other conditions of CauCA hold, then, the causal variables $z_i$ are identifiable. For real-valued $\theta_j$, non-delta priors also imply Assum. 3 almost everywhere.*

Lem. 5 says that when the interventional discrepancy condition is satisfied, then a change in $p(\theta)$ must have occurred. This provides a sufficient criterion to determine when multi-environment data enables representation identification. However, as Assum. 3 is formulated as an almost everywhere condition, the reverse does not necessarily hold—e.g., for discrete RVs such as when $\theta_j$ follows a Bernoulli distribution (Rem. 1). Thus, we prove the reverse for real-valued $\theta_j$.

**Towards the simultaneous identifiability of S, Z, and the DAG.** We finish our discussion of IEM by illustrating how the joint treatment of structure and representation identifiability can be possible with less environments than the two separate problems. As we have shown in § 3.1, it is possible to identify both sources and causal variables by two separate variability conditions (Liu et al., 2024). However, as Assum. 1 requires variability of the structural assignments $g_i$, it cannot be fulfilled by exchangeable sources, at least not for an ANM. Thus, we consider the most general identifiability result in CRL by (Jin & Syrgkanis, 2023), which requires $\dim \mathbf{Z}$ single-node non-degenerate (in the sense of Assum. 3) soft interventions for generic nonparametric SEMs—i.e., when interventions on the exogenous variables change the observational density almost everywhere. Further restricting the sources to first-order conditional exponential family distributions, adding one more intervention can satisfy Assum. 2. Thus, we sidestep the requirement of having $(\dim \mathbf{Z} + 1)$ different environments for ICA, and another $\dim \mathbf{Z}$ for CRL. Namely, by Lem. 5, we know that when Assum. 3 holds, then the parameter priors are non-degenerate. Then, by Lem. 3, Assum. 2 also holds. Thus (proof is in Appx. A.6):

**Lemma 6.** *[Simultaneous identifiability via generic non-degenerate source priors] Provided the assumptions of (Jin & Syrgkanis, 2023, Thm. 4) hold with the restriction of the source variables' density belonging to the exponential family of order one, and assuming that the nonparametric structural assignments are generic such that single-node soft interventions on each $S_i$ satisfy Assum. 3, then* $(\dim \mathbf{Z} + 1)$ *interventions can provide exchangeable data sufficient for the simultaneous identification of both exogenous and causal variables (and also the DAG)—as opposed to* $(2 \dim \mathbf{Z} + 1)$*, where* $\dim \mathbf{Z}$ *separate environments are used for CRL and another* $(\dim \mathbf{Z} + 1)$ *for ICA.*

## 4  DISCUSSION AND FUTURE DIRECTIONS

Our work unifies several Causal Discovery (CD), Independent Component Analysis (ICA), and Causal Representation Learning (CRL) methods with the lens of exchangeability. Next, we answer the question:

*What do we gain from the Identifiable Exchangeable Mechanisms (IEM) framework?*

The motivation of introducing IEM is to provide a unified model that eases understanding and discovery of the synergies between representation and structure identifiability. Our work leverages IEM to relax conditions of general exchangeability to cause and mechanism variability for enabling both structure and representation identifiability. Exchangeability can also model both the passive notion of data variability posited in the ICA literature (e.g., Assum. 2) and the active, agency-based notion of diverse interventions (e.g., Assum. 3). We provide a detailed comparison of the assumptions in both fields in Tab. 1 in Appx. D. By interpreting the variability in ICA as coming from interventions on the exogenous variables, IEM explains why ICA can allow for causal inferences. Namely, assuming that the observations correspond to the causal variables and using ICA to recover the source (exogenous) variables, we can infer the causal graph depending on the identifiability class, as shown by Reizinger et al. (2023).

In the following, we show how exchangeability can model i.i.d., out-of-distribution (OOD) and interventional distributions (§ 4.1), discuss the general conditions that allow for identifiability in the IEM setting (§ 4.2), and detail three additional directions where we believe IEM can open up new possibilities (§§ 4.3 to 4.5).

### 4.1  EXCHANGEABILITY FOR MODELING I.I.D., OOD, AND INTERVENTIONAL DATA

By de Finetti's theorem (de Finetti, 1931), the joint distribution of exchangeable data can be represented as a mixture of i.i.d. distributions $p(x_i|\theta)$, where $\theta$ is drawn from a prior distribution (2). In the special case of $p(\theta) = \delta_{\theta_0}$ we get i.i.d. samples. Guo et al. (2024b) studies how intervention propagates in an exchangeable sequence. Here we note that exchangeability may be a natural choice for modelling OOD and interventional data. For example, when we assume access to multiple environments—where each environment has a distinct parameter drawn from $p(\theta)$: OOD and interventions can be analogusly modelled as a shift in $\theta$, i.e., data in a novel or intervened environment is drawn from $p(x \mid \theta_1)$ instead of $p(x \mid \theta_0)$, where $\theta_1 \neq \theta_0$ (cf. the intution in § 3.1 for an example).

### 4.2  GENERAL CONDITIONS FOR IDENTIFIABILITY IN THE IEM SETTING

When introducing IEM, we focused on exchangeability as a driving factor for structure and representation identifiability. However, theoretical guarantees usually require further assumptions. Here we discuss the general set of assumptions required for identifiabiltiy of the causal structure, the causal variables, and the exogenous (source) variables. We review such assumptions in Tab. 1 in Appx. D. In the case of exchangeable data, we can characterize the best achievable identifiability results as:

**Causal structure (DAG).**  Observed causal variables under faithfulness and cause or mechanism variability are necessary and sufficient to identify the DAG (Thm. 2).

**Causal variables (Z).**  Assuming independent exogenous variables and a diffeomorphic mixing function is sufficient to identify the causal variables up to elementwise nonlinear transformations when we have access to $\dim \mathbf{Z}$ single-node soft interventions with unknown targets (Jin & Syrgkanis, 2023).

**Exogenous (source) variables (S).**  Having exchangeable sources and a surjective feature extractor are sufficient to achieve identifiability up to element-wise nonlinear transformations if the feature extractor is either positive or is augmented by squared features (Khemakhem et al., 2020b).

### 4.3  IDENTIFYING COMPONENTS OF CAUSAL MECHANISMS

Causal mechanisms are composed of exogenous variables and structural assignments. CdF proves the existence of a statistically independent latent variable per causal mechanism. Lem. 2 shows that for ANMs, such latent RVs can be decomposed into separate variables controlling exogenous variables and structural assignments. Wang et al. (2018), for example, performs multi-environment CD via changing only the weights in the linear SEM across environments, which corresponds to changing

the mechanism parameters $\theta^g$. Liu et al. (2024) showed how changing both parameters leads to latent source and causal structure identification. This suggests that partitioning the CdF parameters into mechanism and source parameters can be beneficial to identifying individual components of causal mechanisms.

### 4.4 CAUSE AND MECHANISM VARIABILITY: POTENTIAL GAPS AND FUTURE DIRECTIONS

We relax the assumptions for bivariate CD (§ 3.2) by noticing that changing only either cause or mechanism leads to identifiability, which we term cause and mechanism variability. We further showed with TCL how ICA methods—which usually belong to the cause variability category—can be equivalently extended to mechanism variability (Lem. 4). This dual formulation, though mathematically equivalent, presents new opportunities in practice. Existing work in the ICA literature have focused on identification through variation in the sources with a single deterministic mixing function $f : \mathbf{s} \to \mathbf{o}$, where functional constraints are used for identifiability (Gresele et al., 2021; Lachapelle et al., 2023; Brady et al., 2023; Wiedemer et al., 2023a;b). Multi-view ICA (Gresele et al., 2019), on the other hand, might be related to mechanism variability—we leave investigating this connection to future work.

### 4.5 CHARACTERIZING DEGREES OF NON–I.I.D. DATA

Existing work have developed multiple criteria to characterize non-i.i.d. data from out-of-distribution (Quionero-Candela et al., 2009; Schölkopf et al., 2012; Arjovsky et al., 2020) to out-of-variable (Guo et al.) generalization. Here we assay common criteria for identifiability and highlight potential gaps. Often identification conditions are descriptive with no clear practical guidance in quantifying how and when to induce non-i.i.d. data. Metric computation is also difficult in practice.

**Rank conditions** such as Assum. 2 in TCL (Hyvarinen & Morioka, 2016), our running example for ICA, uses a rank-condition to prove identifiability. Assum. 2 expresses that the multi-environment data is non–i.i.d. However, full-rank matrices can differ to a large extent, e.g., by their condition number, which affects numerical stability, thus, matters in practice (Rajendran et al., 2023). We expect that the condition number could be used to develop bounds for the required sample sizes in practice—an aspect generally missing from the identifiability literature, as most works assume access to infinite samples, with the exception of (Lyu & Fu, 2022).

**Derivative conditions** such as interventional discrepancy (Wendong et al., 2023) require that between environments, there is a non-trivial (i.e., non-zero measure) shift between the causal mechanisms—i.e., the data is not i.i.d. The similarity between interventional discrepancy and the derivative-based condition on the structural assignments in (Liu et al., 2024) (Assum. 1) also has an interesting interpretation: Liu et al. (2024) does not require interventional data *per se*, only non–i.i.d. data that is akin to being generated by a SEM that was intervened on. This assumption is similar to the concepts of Mendelian randomization (Didelez & Sheehan, 2007) or natural experiments (Angrist & Krueger, 1991; Imbens & Angrist, 1994), which assume that an intervention not controlled by the experimenter (but by, e.g., genetic mutations) provides sufficiently diverse data.

**Mechanism shift-based conditions** quantify the number of shifted causal mechanisms. The distribution shift perspective was already present in, e.g., (Zhang et al., 2015; Arjovsky et al., 2020). Perry et al. (2022) explore the SMS hypothesis (Schölkopf, 2019), postulating that domain shifts are due to a sparse change in the set of mechanisms. Their Mechanism Shift Score (MSS) counts the number of changing conditionals, which is minimal for the true DAG. Richens & Everitt (2024) characterize mechanism shifts for causal agents solving decision tasks. Their condition posits that the agent's optimal policy should change when the causal mechanisms shift.

## 5 CONCLUSION

We introduced Identifiable Exchangeable Mechanisms (IEM), a unifying framework that captures a common theme between causal discovery, representation learning, and causal representation learning: access to exchangeable non–i.i.d. data. We showed how particular causal structure and representation identifiability results can be reframed in IEM as exchangeability conditions, from the Causal de Finetti theorem through auxiliary-variable Independent Component Analysis and Causal Component Analysis. Our unified model also led to new insights: we introduced cause and mechanism variability as a special case of exchangeable but not-i.i.d. data, which led us to provide relaxed necessary and sufficient conditions for causal structure identification (Thm. 2), and to formulate identifiability results for mechanism variability-based time-contrastive learning (Lem. 4) We acknowledge that our unified framework might not incorporate all identifiable methods. However, by providing a formal connection between the mostly separately advancing fields of causality and representation learning, more synergies and new results can be developed, just as Thm. 2 and Lem. 4. This, we hope, will inspire further research to investigate the formal connection between these fields.

ACKNOWLEDGMENTS

The authors thank Luigi Gresele and anonymous reviewers for insightful comments and discussions. This work was supported by a Turing AI World-Leading Researcher Fellowship G111021. Patrik Reizinger acknowledges his membership in the European Laboratory for Learning and Intelligent Systems (ELLIS) PhD program and thanks the International Max Planck Research School for Intelligent Systems (IMPRS-IS) for its support. This work was supported by the German Federal Ministry of Education and Research (BMBF): Tübingen AI Center, FKZ: 01IS18039A. Wieland Brendel acknowledges financial support via an Emmy Noether Grant funded by the German Research Foundation (DFG) under grant no. BR 6382/1-1 and via the Open Philantropy Foundation funded by the Good Ventures Foundation. Wieland Brendel is a member of the Machine Learning Cluster of Excellence, EXC number 2064/1 – Project number 390727645. This research utilized compute resources at the Tübingen Machine Learning Cloud, DFG FKZ INST 37/1057-1 FUGG.

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

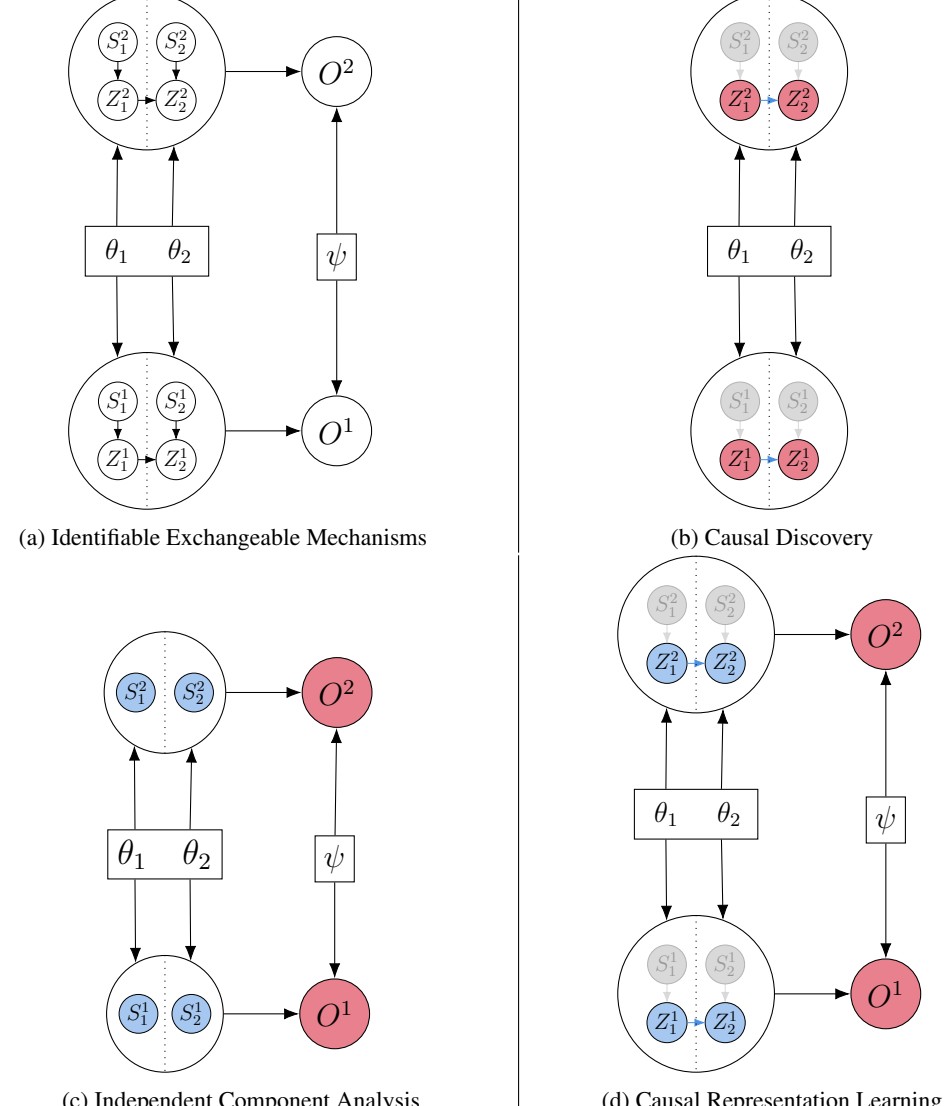

(a) Identifiable Exchangeable Mechanisms

(b) Causal Discovery

(c) Independent Component Analysis

(d) Causal Representation Learning

Figure 4: **Identifiable Exchangeable Mechanisms (IEM)–A unified model for structure and representation identifiability:** Here we show that exchangeable but non-i.i.d. data enables identification in key methods across Causal Discovery (CD), Independent Component Analysis (ICA), and Causal Representation Learning (CRL). The graphical model in Fig. 1a shows the IEM model, which subsumes Causal Discovery (CD) (§ 3.2), Independent Component Analysis (ICA) (§ 3.3), and Causal Representation Learning (CRL) (§ 3.4). $S$ denotes latent, $Z$ causal, and $O$ observed variables with corresponding latent parameters $\boldsymbol{\theta}, \psi$, superscripts denote different samples. Red denotes observed/known quantities, blue stands for target quantities, and gray illustrates components that are *not* explicitly modeled in a particular paradigm. $\theta_i$ are latent variables controlling separate probabilistic mechanisms, indicated by dotted vertical lines. **CD** (Fig. 1b) corresponds to the left-most layer of IEM, focusing on the study of cause-effect relationships between observed causal variables; **ICA** (Fig. 1c) infers source variables from observations, but without causal connections in the left-most layer of IEM; **CRL** (Fig. 1d) shares the most similar structure with IEM, as it has both layers, including the intermediate causal representations

# A PROOFS AND EXTENDED THEORY

## A.1 CAUSE/MECHANISM VARIABILITY FOR BIVARIATE CD: THM. 2

**Theorem 2.** *[Cause/mechanism variability is necessary and sufficient for bivariate CD] Given a sequence of bivariate pairs $\{X^n, Y^n\}_{n \in \mathbb{N}}$ such that for any $N \in \mathbb{N}$, the joint distribution can be represented as:*

- $X \rightarrow Y$: $p(x^1, y^1, \ldots, x^N, y^N) = \int_\theta \int_\psi \prod_n p(y^n | x^n, \psi) p(x^n | \theta) p(\theta) p(\psi) d\theta d\psi$
- $X \leftarrow Y$: $p(x^1, y^1, \ldots, x^N, y^N) = \int_\theta \int_\psi \prod_n p(x^n | y^n, \theta) p(y^n | \psi) p(\psi) p(\theta) d\theta d\psi$

*Then the causal direction between variables $X, Y$ can still be distinguished when:*

1. *either only $p(\theta) = \delta_{\theta_0}(\theta)$ for some constant $\theta_0$ or only $p(\psi) = \delta_{\psi_0}(\psi)$ for some constant $\psi_0$ (but not both). Fig. 2b and Fig. 2c show the Markov structure of such factorizations.*
2. *the distribution of $P$ is faithful (Defn. 4) w.r.t. Fig. 2b or Fig. 2c.*

*Proof.* The impossibility of both mechanisms being degenerate (i.e., the i.i.d. case) is well-known (Pearl, 2009a). For distributions that are Markov and faithful to Fig. 2a, Fig. 2b and Fig. 2c, one can differentiate the causal direction through checking $Y^1 \perp X^2 | X^1$, $X^1 \perp Y^2 | Y^1$ and $X^1 \perp Y^1$. One can observe $Y^1 \perp X^2 | X^1$ only holds in Fig. 5a and fails at Fig. 5b. $\qquad \square$

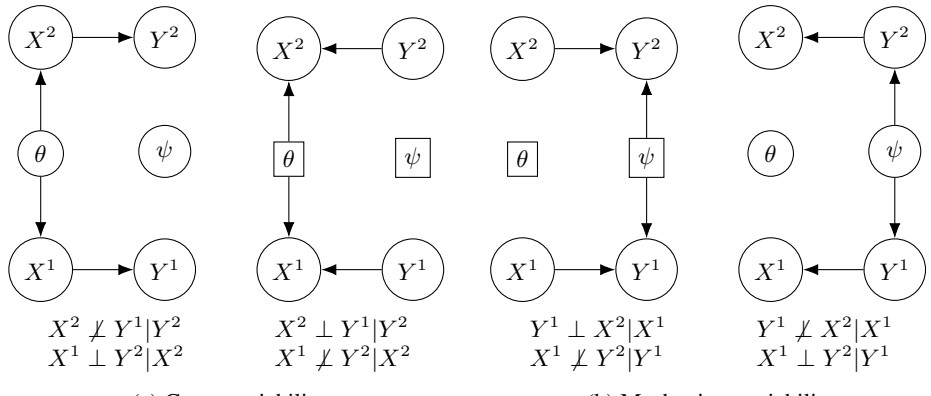

$$X^2 \not\perp Y^1 | Y^2 \qquad X^2 \perp Y^1 | Y^2 \qquad Y^1 \perp X^2 | X^1 \qquad Y^1 \not\perp X^2 | X^1$$
$$X^1 \perp Y^2 | X^2 \qquad X^1 \not\perp Y^2 | X^2 \qquad X^1 \not\perp Y^2 | Y^1 \qquad X^1 \perp Y^2 | Y^1$$

(a) Cause variability            (b) Mechanism variability

Figure 5: We show that the richness argument in CdF (Guo et al., 2024a) can be realized, in the bivariate case, via either only varying the prior of the causes' parameters $\theta$ (Fig. 5a) or the prior of the mechanism' parameters $\psi$ (Fig. 5b). That is, it is not necessary to have rich priors for both $\theta, \psi$

## A.2 EXCHANGEABILITY IN TCL: LEM. 3

**Lemma 3.** *[TCL is identifiable due to exchangeable non–i.i.d. sources] The sufficient variability condition in TCL corresponds to cause variability, i.e., exchangeable non–i.i.d. source variables with a fixed mixing function, which leads to the identifiability of the latent sources.*

*Proof.* Latent variables violating the sufficient variability (Assum. 2) condition in nonlinear ICA imply that those variables are i.i.d.; thus, if more than one latent variables violate this condition, then they become non-identifiable; thus, making non-delta priors a necessary condition for identifiability of factorizing priors (assuming that no further constraints can be applied, e.g., on the function class). An important fact for the proof is that *the parameters $\theta_i$ are continuous RVs*, as they parametrize an exponential family distribution. That is, their support has infinitely many distinct values, each with a probability of zero—this will be important to reason about the zero-measure of edge cases where two parameters happen to be "tuned" to each other, violating the sufficient variability condition.

The full rank condition of the matrix **L** means that $\forall i \neq j$ environment indices for two rows in matrix **L**

$$\boldsymbol{\theta}^i - \boldsymbol{\theta}^0 \neq c \cdot \left( \boldsymbol{\theta}^j - \boldsymbol{\theta}^0 \right); c > 0 \qquad (12)$$

$\Longleftarrow$ *: Non-delta priors imply sufficient variability.* Assume a real-valued RV $\theta_i$ with corresponding non-delta parameter prior $p(\theta_i)$. Then, outside of a zero-measure set, the sufficient variability condition holds. Assume that $\forall i : p(\theta_i) \neq \delta(\theta_i - \theta_{i0})$. Then, as all $\theta_i$ are real RVs, their support has

infinitely many values. Thus, the probability of (12) being violated (by setting both sides to be equal and solving for $\boldsymbol{\theta}^i$) is zero:

$$\Pr\left[\boldsymbol{\theta}^i = c \cdot \boldsymbol{\theta}^j - (c-1)\boldsymbol{\theta}^0\right] = 0, \tag{13}$$

from which it follows that there exists such $\boldsymbol{\theta}^i, \boldsymbol{\theta}^j$ where (12) holds, implying that $\mathbf{L}$ has full rank. Thus, Assum. 2 holds.

$\implies$ : *Sufficient variability implies non-delta priors.* When $\text{rank}(\mathbf{L}) = n$, then none of $p(\theta_i)$ is delta. $\text{rank}\mathbf{L} = n = \dim \mathbf{s}$ means that $\forall i \neq j$ environment indices for two rows in matrix $\mathbf{L}$ (12) holds. That is, we can construct $\mathbf{L}$ such that any two rows are linearly independent[3]. If for coordinate $k$, $\theta_k^e = \theta_k = const$, then $\mathbf{L}$ cannot have full column rank. Since $\theta_k$ cannot be constant for all $k$, this requires that $p(\theta_k)$ is non-delta. $\square$

### A.3 Exchangeability in GCL (Hyvarinen et al., 2019): extending Lem. 3

Hyvarinen et al. (2019) proposed a generalization of TCL (and other auxiliary-variable ICA methods), called GCL. GCL uses a more general conditional distribution, it only assumes that assumes that the conditional log-density $\log p(\mathbf{s}|u)$ is a sum of components $q_i(s_i, u)$:

$$\log p(\mathbf{s}|u) = \sum_i q_i(s_i, u) \tag{14}$$

For this generalized model, they define the following variability condition:

**Assumption 4** (Assumption of Variability). *For any $\mathbf{y} \in \mathbb{R}^n$ (used as a drop-in replacement for the sources $\mathbf{s}$), there exist $2n + 1$ values for the auxiliary variable $\mathbf{u}$, denoted by $\mathbf{u}_j, j = 0 \ldots 2n$ such that the $2n$ vectors in $\mathbb{R}^{2n}$ given by*

$$\left(\mathbf{w}\left(\mathbf{y}, \mathbf{u}_1\right) - \mathbf{w}\left(\mathbf{y}, \mathbf{u}_0\right)\right), \left(\mathbf{w}\left(\mathbf{y}, \mathbf{u}_2\right) - \mathbf{w}\left(\mathbf{y}, \mathbf{u}_0\right)\right) \ldots, \left(\mathbf{w}\left(\mathbf{y}, \mathbf{u}_{2n}\right) - \mathbf{w}\left(\mathbf{y}, \mathbf{u}_0\right)\right)$$

*with*

$$\mathbf{w}(\mathbf{y}, \mathbf{u}) = \left(\frac{\partial q_1\left(y_1, \mathbf{u}\right)}{\partial y_1}, \ldots, \frac{\partial q_n\left(y_n, \mathbf{u}\right)}{\partial y_n}, \quad \frac{\partial^2 q_1\left(y_1, \mathbf{u}\right)}{\partial y_1^2}, \ldots, \frac{\partial^2 q_n\left(y_n, \mathbf{u}\right)}{\partial y_n^2}\right)$$

*are linearly independent.*

Assum. 4 puts a constraint on the components of the first- and second derivatives of the functions constituting the conditional log-density of the source/latent variables, conditioned on the auxiliary variable $\mathbf{u}$. As the authors write: *"[Assum. 4] is basically saying that the auxiliary variable must have a sufficiently strong and diverse effect on the distributions of the independent components."*

We focus on a special case, which assumes that the source conditional log-densities $q_i(s_i, u)$ are conditionally exponential, i.e.:

$$q_i\left(s_i, u\right) = \sum_{j=1}^{k} \left[\tilde{q}_{ij}\left(s_i\right) \theta_{ij}(u)\right] - \log N_i(u) + \log Q_i(s_i), \tag{15}$$

where $k$ is the order of the exponential family, $N_i$ is the normalizing constant, $Q_i$ the base measure, $\tilde{q}_i$ is the sufficient statistics, and the modulation parameters $\theta_i := \theta_i(u)$ depend on $u$. In this case, Assum. 4 becomes similar to Assum. 2, but the modulation parameter matrix now has $(E-1) \times nk$ dimensions, where the rows are:

$$[\mathbf{L}]_{j:} = (\boldsymbol{\theta}^j - \boldsymbol{\theta}^0)^\top \tag{16}$$

$$\boldsymbol{\theta}^j = \left[\theta_{11}^j, \ldots, \theta_{nk}^j\right]. \tag{17}$$

In this case, we can generalize Lem. 3 to GCL:

**Corollary 1** (GCL with conditionally exponential family sources is identifiable due to exchangeable non–i.i.d. sources). *The sufficient variability condition in GCL with conditionally exponential family sources corresponds to cause variability, i.e., exchangeable non–i.i.d. source variables with fixed mixing function, which leads to the identifiability of the latent sources.*

*Proof.* The proof follows from the proof of Lem. 3, the only difference is that for each source $s_i$, there are $k$ sufficient statistics $\tilde{q}_{ik}$ and modulation parameters $\theta_{ik}(u)$. Thus, the modulation parameter matrix $\mathbf{L}$ (Assum. 2) will be $[(E-1) \times nk]-$dimensional where $n = \dim \mathbf{s}$. $\square$

---

[3]Note that $\theta_i$ can be correlated, as Hyvarinen et al. (2019) pointed out in the proof of their Thm. 2

## A.4 DUALITY OF CAUSE AND MECHANISM VARIABILITY FOR TCL: LEM. 4

**Lemma 4.** *[Duality of cause and mechanism variability for TCL] For a given deterministic mixing function $f : \mathbf{s} \to \mathbf{o}$ and conditionally factorizing (non-stationary) latent sources $p(\mathbf{s}|u) = \prod_i p_i(s_i|u)$ fulfilling the sufficient variability of TCL, there exists an equivalent setup with stationary (i.i.d.) sources $p(\mathbf{s}) = \prod_i p_i(s_i)$ with stochastic functions $\hat{f} = f \circ g : \mathbf{s} \to \mathbf{o}$, where $g = g(u)$ and each component $g_i$ is defined as an element-wise function such that the pushforward of $p_i(s_i)$ by $g_i$ equals $p_i(s_i|u)$, i.e., $g_{i*}p_i(s_i) = p_i(s_i|u)$. Then, $g_{i*}p_i(s_i)$ fulfils the same variability condition; thus, the same identifiability result applies.*

*Proof.* The proof follows from the observation that it is a modelling choice which component provides the source of non-stationarity. That is, we can incorporate the transformation of the source variables into the source distribution (cause variability, as TCL does) or we can think of that as stochasticity in the mixing function (mechanism variability). □

## A.5 EXCHANGEABILITY IN CAUCA: LEM. 5

**Lemma 5.** *[Non-delta priors in the causal mechanisms can enable identifiable CRL]*
*If the interventional discrepancy condition Assum. 3 holds, then the parameter priors in (10) cannot equal a delta distribution, i.e., $p(\theta_j) \neq \delta_{\theta_{j0}}(\theta_j)$; thus, if the other conditions of CauCA hold, then, the causal variables $z_i$ are identifiable. For real-valued $\theta_j$, non-delta priors also imply Assum. 3 almost everywhere.*

*Proof.* We define each mechanism $p(z_i|\mathbf{Pa}(z_i))$ as

$$p(z_i|\mathbf{Pa}(z_i)) = \int_{\theta_i} p(z_i|\mathbf{Pa}(z_i), \theta_i)p(\theta_i)d\theta_i \tag{18}$$

Thus, the observational and interventional mechanisms, respectively, are:

$$p_i = p(z_i|\mathbf{Pa}(z_i), \theta_i = \theta_i^0) \tag{19}$$

$$\tilde{p}_i = p(z_i|\mathbf{Pa}(z_i), \theta_i = \tilde{\theta}_i) \tag{20}$$

That is, each intervention corresponds to a specific parameter value $\theta_i$ (which exist by the CdF theorem (Thm. 1). Thus, (18) is akin to mixtures of interventions (Richens & Everitt, 2024, Defn. 3).
$\implies$ : *Interventional discrepancy implies non-delta priors.*
We prove this direction by contradiction. Assume that Assum. 3 is fulfilled and $p(\theta_i) = \delta_{\theta_{i0}}$. Then $\tilde{\theta}_i = \theta_i$, so Assum. 3 cannot hold.
$\impliedby$ : *Non-delta priors for real-valued parameters imply interventional discrepancy.*
If $\theta_i$ is real-valued, then the following probability is zero:

$$\Pr\left[\tilde{\theta}_i = \theta_i^0\right] = 0, \tag{21}$$

thus, there must exist $\tilde{\theta}_i \neq \theta_i^0$, and this inequality holds almost everywhere since:

$$\Pr\left[\tilde{\theta}_i \neq \theta_i^0\right] = 1 - \Pr\left[\tilde{\theta}_i = \theta_i^0\right] = 1 - 0 = 1, \tag{22}$$

□

**Remark 1** (Non-delta priors do not always imply interventional discrepancy almost everywhere)**.**
*When the parameter priors $p(\theta_i)$ are not a delta distribution, then, barring the case when sampling the interventional mechanism parameter $\tilde{\theta}_i$ yields the same $\theta_i^0$, then the distributions $p_i$ and $\tilde{p}_i$ would differ. However, this is not necessarily a zero-measure event, e.g., when $p(\theta)$ has a Bernoulli distribution with parameter $1/2$. Thus, Assum. 3 cannot hold almost everywhere without further restrictions.*

## A.6 EXCHANGEABILITY AND THE GENERICITY CONDITION FROM VON KÜGELGEN ET AL. (2023) AND JIN & SYRGKANIS (2023)

von Kügelgen et al. (2023) extends CauCA Wendong et al. (2023) by providing identifiability proofs for CRL without parametric assumptions on the function class. Their assumption (von Kügelgen et al., 2023, (A3') in Thm. 3.4) is stronger than Assum. 3 as it requires that two interventional densities differ *everywhere*—or that the observational and one interventional density differ (von Kügelgen

et al., 2023, (A3) in Thm. 3.2). Furthermore, (von Kügelgen et al., 2023, (A4) in Thm. 3.2) excludes the pathogological case of fine-tuned densities (thus the name *genericity condition*)—this might be thought to be an analog of faithfulness Defn. 4.

Jin & Syrgkanis (2023) leverages a similar assumption as Assum. 3 in their (Jin & Syrgkanis, 2023, Def. 6). They strengthen the nonparametric identifiability results of von Kügelgen et al. (2023) by showing that $\dim \mathbf{Z}$ single-node soft interventions with unknown targets are sufficient to identify the causal variables.

**Towards identifying the exogenous variables in CRL.**   Both von Kügelgen et al. (2023); Jin & Syrgkanis (2023) derive identifiability results for the *causal variables* from interventional data. However, they do not make claims about the exogenous variables. Based on the insights of our unified model, IEM, provides, we investigate whether there is an identifiability proof that encompasses the whole hierarchy.

Consider that the mechanism $p(Z_i|\mathbf{Pa}(Z_i))$ can be intervened upon by changing how the other causal variables $Z_j \in \mathbf{Pa}(Z_i)$ affect $Z_i$. Alternatively, $p(Z_i|\mathbf{Pa}(Z_i))$ can also change by modifying the distribution of the corresponding exogenous variable $S_i$—note that this corresponds to a one-node soft intervention. Thus, if the other assumptions of Jin & Syrgkanis (2023) holds, we can say that: single-node soft interventions on $S_i$ can satisfy the genericity condition of Jin & Syrgkanis (2023).

To reason about the identifiability of the exogenous variables, we need the variability of their distribution across the available environments. Our summary of the assumptions in Tab. 1 suggests that with sufficiently many environments, it should be possible to identify the exogenous variables as well. If the only assumption we make is exchangeability, then, following the reasoning of Khemakhem et al. (2020b), we might need $\dim \mathbf{Z} + 1$ additional environments ($2 \dim \mathbf{Z} + 1$ in total). If we further restrict the source distributions to belong to the exponential family, then we can apply TCL (and linear ICA on top) to identify the source variables. Thus, we can state:

**Lemma 6.** *[Simultaneous identifiability via generic non-degenerate source priors] Provided the assumptions of (Jin & Syrgkanis, 2023, Thm. 4) hold with the restriction of the source variables' density belonging to the exponential family of order one, and assuming that the nonparametric structural assignments are generic such that single-node soft interventions on each $S_i$ satisfy Assum. 3, then $(\dim \mathbf{Z} + 1)$ interventions can provide exchangeable data sufficient for the simultaneous identification of both exogenous and causal variables (and also the DAG)—as opposed to $(2 \dim \mathbf{Z} + 1)$, where $\dim \mathbf{Z}$ separate environments are used for CRL and another $(\dim \mathbf{Z} + 1)$ for ICA.*

*Proof.* Our goal is to prove that performing CRL can lead to the additional identifiability of the exogenous sources, with a negligible overhead in terms of assumptions on the data, compared to performing only CRL. Also, we aim to show that the joint identifiability requires less data (less environments) than performing both tasks separately. We start by assuming that (Jin & Syrgkanis, 2023, Thm. 4) holds, which implies Assum. 3. By Lem. 5, we know that when Assum. 3 holds, then the parameter priors are non-degenerate. We assumed that the single-node soft interventions only affect $S_i$. Following the reasonings of von Kügelgen et al. (2023); Jin & Syrgkanis (2023), w.l.o.g., if the structural assignments in the nonparametric SEM are not fine-tuned (i.e., they are generic), then Assum. 3 should hold. Then, as the source distributions are exchangeable, we can apply Lem. 3, which states that Assum. 2 also holds. Thus, we can identify the exogenous variables as well, concluding the proof. □

We leave it to future work to investigate whether identifying both causal and exogenous variables is possible from fewer environments. Nonetheless, we believe that this example shows the potential advantage of the IEM framework for providing new identifiability results.

### A.7   EXCHANGEABILITY IN THE UNIFIED MODEL: LEM. 1

**Interventional discrepancy and the derivative condition of (Liu et al., 2024)**   The identifiability result of Liu et al. (2024) combines the results from ICA and CRL. As they use TCL to learn the latent sources, we can apply Lem. 3. To see how the causal variables and the edges between them can also be learned, we first relate the derivative condition on the structural assignments to the interventional discrepancy condition of Wendong et al. (2023) (Assum. 3).

Assum. 1 requires access to a set of environments (indexed by auxiliary variable $u$), such that for each parent $z_j \in \mathbf{Pa}(z_i)$ node, there is an environment, where the edge $z_j \to z_i$ is blocked. To relate Assum. 1 to the interventional discrepancy Assum. 3, we recall that Wendong et al. (2023) note that for perfect interventions, the conditioning on the parents for the interventional density in Assum. 3 disappears. Thus, we interpret Assum. 1 as "emulating" perfect interventions for each $z_i$. By this, we

mean that we need data from such environments, where the structural assignments change *as if* a perfect intervention is carried out to remove the $z_j \rightarrow z_i$ edge.

**Lemma 1.** *[Identifiable Latent Neural Causal Models are identifiable with exchangeable sources and mechanisms] The model of Liu et al. (2024) (Fig. 1a) identifies both the latent sources* **s** *and the causal variables* **z** *(including the graph), by the variability of* **s** *via a non-delta prior over $\theta^s$ and by the variability of the structural assignments via $\theta^g$.*

*Proof.* As the authors rely on TCL and a form of the interventional discrepancy Assum. 3, the proof follows from Lem. 3 and Lem. 5. □

### A.8 INDEPENDENT SOURCE AND STRUCTURAL ASSIGNMENT CDF PARAMETERS IN ANMS: LEM. 2

**Lemma 2.** *[Independent source and structural assignment CdF parameters for ANMs] In the setting of Liu et al. (2024), where the SEM is an ANM, the CdF parameters for the sources, $\boldsymbol{\theta}^s$, and the structural assignments, $\boldsymbol{\theta}^g$, are independent, i.e. $p(\boldsymbol{\theta}^g, \boldsymbol{\theta}^s) = p(\boldsymbol{\theta^g})p(\boldsymbol{\theta^s})$.*

*Proof.* In the model of Liu et al. (2024), the two identifiability results impose two non-i.i.d. requirements: to identify the latent sources, a sufficient variability condition from TCL is required (cf. the generalized version in Assum. 4), whereas for CRL, a derivative-based condition on the mechanisms (akin to Assum. 3) is required. As the SEM is an ANM in this case, defined by $z_i := g_i(\mathbf{Pa}(z_i)) + s_i$, and the exogenous variables are assumed to be independent from $g_i(\mathbf{Pa}(z_i))$. Thus, it is impossible (assuming faithfulness) that a change in $\theta_i^s$ would change $\theta_i^g$; otherwise, $g_i(\mathbf{Pa}(z_i))$ and $s_i$ would be dependent. That is, their parameters are independent. □

## B DEFINITIONS

**Definition 2** (*d*-separation (adapted from Defn. 6.1 in (Peters et al., 2018))). *Given a DAG $\mathcal{G}$, the disjoint subsets of nodes $A$ and $B$ are $d$-separated by a third (also disjoint) subset $S$ if every path between nodes in $A$ and $B$ is blocked by $S$. We then write*

$$A \perp_{\mathcal{G}} B \mid S.$$

**Definition 3** (Global Markov property (adapted from Defn. 6.21(i) in (Peters et al., 2018))). *Given a DAG $\mathcal{G}$ and a joint distribution $P$, $P$ satisfies the global Markov property w.r.t. the $\mathcal{G}$ if*

$$A \perp_{\mathcal{G}} B|C \Rightarrow A \perp B|C \tag{23}$$

*for all disjoint vertex sets $A, B, C$ (the symbol $\perp_{\mathcal{G}}$ denotes $d$-separation, cf. Defn. 2).*

**Definition 4** (Faithfulness (adapted from Defn. 6.33 in (Peters et al., 2018))). *Consider a distribution $P$ and a DAG $\mathcal{G}$. Then, $P$ is faithful to $\mathcal{G}$ if for all disjoint node sets $A, B, C$:*

$$A \perp B \mid C \Rightarrow A \perp_{\mathcal{G}} B \mid C. \tag{24}$$

*That is, if a conditional independence relationship holds in $P$, then the corresponding node sets are $d$-separated in $\mathcal{G}$.*

**Definition 5** (Markov equivalence class of graphs (adapted from Defn. 6.24 in (Peters et al., 2018))). *We denote by $\mathcal{M}(\mathcal{G})$ the set of distributions that are Markovian w.r.t. $\mathcal{G}$ : $\mathcal{M}(\mathcal{G}) := \{P : P$ satisfies the global Markov property w.r.t. $\mathcal{G}\}$. Two DAGs $\mathcal{G}_1$ and $\mathcal{G}_2$ are Markov equivalent if $\mathcal{M}(\mathcal{G}_1) = \mathcal{M}(\mathcal{G}_2)$, i.e.. if and only if $\mathcal{G}_1$ and $\mathcal{G}_2$ satisfy the same set of $d$-separations, which means the Markov condition entails the same set of (conditional) independence conditions. The set of all DAGs that are Markov equivalent to some DAG is called Markov equivalence class of $\mathcal{G}$.*

## C WHY DOES I.I.D. DATA FAIL?

We next assay key results and provide concrete examples to illustrate why i.i.d. data fails to enable identification for both structure and representation learning.

**Example 2** (Bivariate CD is impossible from i.i.d. data (Pearl, 2009b)). *One cannot distinguish $X \rightarrow Y$ from $X \leftarrow Y$ from i.i.d. data as both structures imply identical graphical conditional independence, i.e., $\emptyset$. Thus, bivariate CD is impossible in i.i.d. data without further parametric assumptions.*

Learning disentangled latent factors is also impossible without further parametric assumptions in i.i.d. data (Hyvärinen & Pajunen, 1999; Locatello et al., 2019):

**Example 3** (Gaussian latent factors are not identifiable from i.i.d. data). *Assume independent latents with Gaussian components, i.e. $p(\mathbf{s}) = \prod_i p_i(s_i)$, where $p_i(s_i) = \mathcal{N}\left(\mu_i; \sigma_i^2\right)$. Even if $\forall i, j : \sigma_i^2 \neq \sigma_{j \neq i}^2$, Gaussian sources are not identifiable to their rotational symmetry and the scale-invariance of ICA.*

We show in § 3 how exchangeability unifies the non-i.i.d. conditions (often termed, weak supervision or auxiliary information) in many causal structure and representation identifiability methods.

## D    RELATED WORK

**Identifiable representation learning and ICA.**    Identifiable representation learning aims to learn (low-dimensional) latent variable models (LVMs) from (high-dimensional) observations. The most prevalent family of models is that of Independent Component Analysis (ICA) (Comon, 1994; Hyvarinen et al., 2001), which assumes that the observations are a mixture of *independent* variables via a deterministic mixing function. Identifiability means that the latents can be can be recovered up to indeterminacies (e.g., permutation, element-wise transformations). As this is provably impossible in the nonlinear case without further assumptions (Darmois, 1951; Hyvärinen & Pajunen, 1999; Locatello et al., 2019), recent work has focused *auxiliary-variable* ICA, where the latents are conditionally independent given the auxiliary variable $u$ (Hyvarinen et al., 2019; Gresele et al., 2019; Khemakhem et al., 2020a; Hälvä et al., 2021; Hyvarinen & Morioka, 2017; 2016; Hälvä & Hyvärinen, 2020; Morioka et al., 2021; Monti et al., 2020; Hyvarinen et al., 2010; Klindt et al., 2021; Zimmermann et al., 2021)—despite the latents are not marginally independent, the literature still refers to these models are ICA. Several such methods model multiple environments with an auxiliary variable, which is also known as using ensembles (Eastwood et al., 2023; Kirchhof et al., 2023). We note that some methods make functional assumptions (Shimizu et al., 2006; Hoyer et al., 2008; Zhang & Hyvarinen, 2012; Gresele et al., 2021), but our focus is on the auxiliary-variable methods. Recently, Bizeul et al. (2024) developed a probabilistic model for self-supervised representation learning, including auxiliary-variable ICA methods.

**Causality.**    SEMs model cause-effect relationships between causal variables $z_i$, where each $z_i$ is determined by a deterministic function $g_i(\mathbf{Pa}(z_i))$, where $\mathbf{Pa}(z_i)$ includes all the $z_{j<i}$ causal variables that cause $z_i$ and also the stochastic exogenous variable $x_i$. Learning causal models enables to make more fine-grained (interventional, counterfactual) queries compared to observational data (Pearl, 2009b). CD aims to uncover the graph between the $z_i$ from observing $z_i$. This admits interventional queries. CRL also learns that $z_i$ from high-dimensional observations (Schölkopf et al., 2021). Causal methods need to rely on certain assumptions, either restricting the distribution of the exogenous variables (Kalainathan et al., 2020; Lachapelle et al., 2020; Shimizu et al., 2006; Monti et al., 2020), and/or the function class of the SEM (Shimizu et al., 2006; Zheng et al., 2018; Squires et al., 2023; Montagna et al., 2023b;a; Gresele et al., 2021; Hoyer et al., 2008; Ng et al., 2020; Lachapelle et al., 2020; Annadani et al., 2021; Yang et al., 2021). Concurrently with our work, Yao et al. (2024) provides an invariance-based framework to unify CRL. Their framework can encompass multi-view, multi-environment, and also temporal settings—ouw work focuses on the multi-environment case, but it also includes representation learning and CD.

**Connections between representation learning and causality.**    Causality and identifiability both aim to recover some ground truth structures (latent factors, DAGs, or functional relationships), thus, several works explored possible connections (Reizinger et al., 2023; Morioka & Hyvarinen, 2023; Hyvärinen et al., 2023; Zečević et al., 2021; Richens & Everitt, 2024; Monti et al., 2020). Several methods connected ICA to the SEM model in causality (Gresele et al., 2021; Monti et al., 2020; Shimizu et al., 2006; von Kügelgen et al., 2021; Hyvärinen et al., 2023). An important observation we rely on is that identifiability guarantees from require a notion of non-i.i.d.ness, e.g., both the ICA and the causal literature often relies on the multi-environmental setting.

Table 1: **Mixing assumptions:** $p(\mathbf{s})$ stands for assumptions on the source distribution, $f$ on the mixing function, $\perp$ stands for independence (the superscript $\mathbf{u}$ denotes conditional independence given $\mathbf{u}$), *CEF* for conditional exponential family (the superscript $+$ denotes monotonicity, 2 denotes a CEF of order two), *ING* for independent non-Gaussian (in Jin & Syrgkanis (2023) maximum one Gaussian component allowed, the distributions need to be different; in Zimmermann et al. (2021), the $L^{\alpha}$ metric is such that $\alpha \geq 1, \alpha \neq 2$), *exg.* stands for exchangeability, *AG* for an anistropic Gaussian on the hypersphere, *EnAG* for an ensemble of such Gaussians, *inj.* for injectivity, *surj.* for subjectivity, $C^2$ for diffeomorphism; **SEM assumptions:** $\mathbf{Z}$ stands for assumptions on the causal variables, $g$ on the SEM, $\mathcal{M}$ denotes the Markov assumption, $\mathcal{F}$ faithfulness (or lack thereof), *NP* stands for non-parametric; **Interventional (variability) assumptions:** $\#$ denotes the number of nodes affected by the intervention, *P/S* denotes perfect or soft interventions, the *target* column whether the intervention targets are known, $|E|$ stands for the number of environments ($d = \dim \mathbf{Z}$), $k$ is the order of the exponential family; **Identifiability ambiguities:** DAG denotes identifiability of the causal graph ($\checkmark$ means the DAG is known; $\checkmark$'s come from the result of Reizinger et al. (2023)), $h$ denotes identifiability up to elementwise (non-)linear transformations, $\mathbf{D}$ denotes scaling, $\pi$ permutations, $c$ a constant shift, $\mathbf{O}$ an orthogonal, $\{\mathbf{O}\}$ a block-orthogonal, $\mathbf{A}$ an invertible matrix.

| | Mixing | | SEM | | Interventions | | | | Ident. $Z$ | | | | | Ident. $S$ | | | | |
|---|---|---|---|---|---|---|---|---|---|---|---|---|---|---|---|---|---|---|
| Method | $\mathbf{S}$ | $f$ | $\mathbf{Z}$ | $g$ | $\#$ | type | target | $|E|$ | DAG | $h$ | $\mathbf{D}$ | $\pi$ | $c$ | $\mathbf{A}$ | $h$ | $\mathbf{D}$ | $\pi$ | $c$ |
| Guo et al. (2024a) | | | $\mathcal{F}$ | exg. | 0 | | | | $\checkmark$ | | | | | | | | | |
| Hyvarinen & Morioka (2016) | CEF | $C^2$ | | | - | S | ✗ | $d+1$ | | | | | | $\checkmark$ | $\checkmark$ | $\checkmark$ | $\checkmark$ | $\checkmark$ |
| Hyvarinen & Morioka (2016) | $\text{CEF}^+$ | $C^2$ | | | - | S | ✗ | $d+1$ | $\checkmark$ | | | | | ✗ | $\checkmark$ | $\checkmark$ | $\checkmark$ | $\checkmark$ |
| Hyvarinen et al. (2019) | $\perp^{\mathbf{u}}$ | $C^2$ | | | - | S | ✗ | $2d+1$ | $\checkmark$ | | | | | ✗ | $\checkmark$ | $\checkmark$ | $\checkmark$ | ✗ |
| Hyvarinen et al. (2019) | CEF | $C^2$ | | | - | S | ✗ | $dk+1$ | | | | | | $\checkmark$ | $\checkmark$ | $\checkmark$ | $\checkmark$ | $\checkmark$ |
| Zimmermann et al. (2021) | vMF | $C^2$ | | | $d$ | S | ✗ | 1 | | | | | | $\mathbf{O}$ | ✗ | ✗ | $\checkmark$ | ✗ |
| Zimmermann et al. (2021) | $\mathbb{R}$ | $C^2$ | | | $d$ | S | ✗ | 1 | | | | | | $\checkmark$ | ✗ | ✗ | $\checkmark$ | $\checkmark$ |
| Zimmermann et al. (2021) | ING | $C^2$ | | | $d$ | S | ✗ | 1 | $\checkmark$ | | | | | ✗ | ✗ | $\checkmark$ | $\checkmark$ | ✗ |
| Rusak et al. (2024) | AG | $C^2$ | | | $d$ | S | ✗ | 1 | | | | | | $\{\mathbf{O}\}$ | ✗ | ✗ | $\checkmark$ | ✗ |
| Rusak et al. (2024) | EnAG | $C^2$ | | | - | S | ✗ | $1<$ | $\checkmark$ | | | | | ✗ | ✗ | ✗ | $\checkmark$ | ✗ |
| Khemakhem et al. (2020a) | $\text{CEF}^2$ | inj. | | | - | S | ✗ | $dk+1$ | $\checkmark$ | | | | | ✗ | $\checkmark$ | $\checkmark$ | $\checkmark$ | $\checkmark$ |
| Khemakhem et al. (2020b) | exg. | $\text{surj.}^4$ | | | - | S | ✗ | $2d+1$ | | | | | | $\checkmark$ | ✗ | $\checkmark$ | $\checkmark$ | $\checkmark$ |
| Khemakhem et al. (2020b) | exg. | $\text{surj.}^+$ | | | - | S | ✗ | $2d+1$ | $\checkmark$ | | | | | ✗ | ✗ | $\checkmark$ | $\checkmark$ | $\checkmark$ |
| Khemakhem et al. (2020b) | exg. | $\text{surj.}^2$ | | | - | S | ✗ | $2d+1$ | $\checkmark$ | | | | | ✗ | ✗ | $\checkmark$ | $\checkmark$ | $\checkmark$ |
| Reizinger et al. (2023) | | | | | | | | | $\checkmark$ | | | | | ✗ | $\checkmark$ | $\checkmark$ | $\checkmark$ | $\checkmark$ |
| Wendong et al. (2023) | $\perp$ | $C^2$ | $\mathcal{M}$ | | 1 | S | $\checkmark$ | $d$ | $\checkmark$ | $\checkmark$ | $\checkmark$ | ✗ | $\checkmark$ | | | | | |
| Wendong et al. (2023) | $\perp$ | $C^2$ | $\mathcal{M}$ | | 1 | P | $\checkmark$ | $d$ | $\checkmark$ | ✗ | $\checkmark$ | ✗ | ✗ | | | | | |
| von Kügelgen et al. (2023) | $\perp$ | $C^2$ | $\mathcal{F}$ | NP | 1 | P | ✗ | $2d$ | $\checkmark$ | $\checkmark$ | $\checkmark$ | $\checkmark$ | $\checkmark$ | | | | | |
| Jin & Syrgkanis (2023) | ING | $C^2$ | $\cancel{\mathcal{F}}$ | lin | 1 | S | ✗ | $d^2$ | $\checkmark$ | $\checkmark$ | $\checkmark$ | $\checkmark$ | $\checkmark$ | | | | | |
| Jin & Syrgkanis (2023) | ING | $C^2$ | $\cancel{\mathcal{F}}$ | lin | - | S | ✗ | $d$ | $\checkmark$ | $\checkmark$ | $\checkmark$ | $\checkmark$ | $\checkmark$ | | | | | |
| Jin & Syrgkanis (2023) | $\perp$ | $C^2$ | $\cancel{\mathcal{F}}$ | NP | 1 | S | ✗ | $d$ | $\checkmark$ | $\checkmark$ | $\checkmark$ | $\checkmark$ | $\checkmark$ | | | | | |
| Liu et al. (2024) | $\text{CEF}^2$ | inj. | $\mathcal{F}$ | ANM | 1 | P | ✗ | $2d+1$ | $\checkmark$ | ✗ | $\checkmark$ | ✗ | $\checkmark$ | ✗ | ✗ | $\checkmark$ | $\checkmark$ | $\checkmark$ |

# E  INTUITION AND EXAMPLES FOR CAUSE AND MECHANISM VARIABILITY

**The Sparse Mechanism Shift hypothesis motivates cause and mechanism variability.**   In § 3, we relaxed exchangeability into cause and mechanism variability. In this section, we show that both cause and mechanism variability can be used to describe many real-world scenarios. Intuitively,

*Cause and mechanism variability can be seen as particular realizations of the Sparse Mechanism Shift (SMS) hypothesis (Perry et al., 2022).*

The SMS posits that the causal mechanisms (the factors in the causal Markov factorization) tend to change sparsely, i.e., interventions or distribution shifts can be described by changing a (strict) subset of mechanisms. This is one main argument for the efficiency of causal modelling, as the modularity implies that only parts of the model need to be adapted in case of a distribution shift—in contrast to a non-causal factorization, where the whole learned model needs to be fine-tuned.

---

[4]In ICE-BeeM (Khemakhem et al., 2020b), the assumption is on the feature extractor

Indeed, the SMS hypothesis captures the reasoning behind many works in causality (Gendron et al., 2023; Perry et al., 2022; Lachapelle et al., 2021b; 2022; Schölkopf et al., 2012; Lachapelle et al., 2021a; Ahuja et al., 2022b). Sparse changes have been also connected to causal modeling (Rajendran et al., 2023; von Kügelgen et al., 2021; Fumero et al., 2023; Mansouri et al., 2023; Ahuja et al., 2022a).

## E.1 REAL-WORLD EXAMPLES

In this section, we draw on prior works to provide real-world examples of cause and mechanism variability—for examples of the exchangeable case, we refer the reader to (Guo et al., 2024a). As with any model, we will make certain simplifications, though we aim to convey that the principle of cause and mechanism variability still applies. We will restrict ourselves to the bivariate case, as in Fig. 5. The causal factorization for $X \to Y$ is $p(Y|X, \psi)p(X|\theta)$, where the CdF parameters are $\theta, \psi$. Cause variability means that $p(\psi)$ is a delta distribution, whereas mechanism variability means that $p(\theta)$ is a delta distribution.

### E.1.1 CAUSE VARIABILITY.

**Example 4** (Lung cancer). *Assume that $\theta$ parametrizes the lifestyle, socioeconomic, and environmental factors of people, whereas $\psi$ parametrizes how lung cancer develops. In this case, we can assume that $p(X|\theta)$ differs across cities, whereas the mechanism for developing lung cancer, $p(Y|X, \psi)$ is the same. That is, only $p(\psi)$ is a delta distribution.*

**Example 5** (Altitude and temperature). *Assume that $\theta$ parametrizes the altitude distribution of countries, whereas $\psi$ parametrizes how altitude affects temperature. In this case, we can assume that $p(X|\theta)$ differs across countries, whereas the effect of altitude on temperature $p(Y|X, \psi)$ is the same. That is, only $p(\psi)$ is a delta distribution.*

### E.1.2 MECHANISM VARIABILITY

**Example 6** (Natural experiments). *In natural experiments in economics (Angrist & Krueger, 1991; Imbens & Angrist, 1994), it is possible to select two populations such that we can assume that their distributions are the same, i.e., the corresponding $\theta$ parameter has a delta distribution, whereas the economic situation, parametrized by $\psi$, differs, e.g., by the two cities having different local taxes.*

**Example 7** (Medical diagnoses). *Assume that several people having the same lifestyle, socioeconomic, and environmental status are admitted to the same hospital after food poisoning at a local restaurant. Then, the probability distribution describing the symptoms, parametrized by $\theta$, will have a delta prior, as each person suffers from the same disease. If we assume that multiple doctors are required to diagnose and treat all patients, then we can posit that there will be (slight) differences in their decisions and prescribed treatments, which means that the corresponding parametric mechanism $p(Y|X, \psi)$ for the treatment has a non-delta prior for $\psi$.*

## F EXPERIMENTAL RESULTS: CAUSE AND MECHANISM VARIABILITY FOR CAUSAL DISCOVERY

**Setup.** To demonstrate that both cause and mechanism variability enable causal structure identification, we ran synthetic experiments based on the publicly available repository of the Causal de Finetti paper[5]. We focus on the continuous case, as problems can arise for discrete RVs (e.g., in Lem. 5)—i.e., we follow the protocol described in the *"Bivariate Causal Discovery"* paragraph in (Guo et al., 2024a, Sec. 6). The continuous experiments used in the original CdF paper consider the bivariate case, which we follow to be comparable. The only change in the evaluation protocol is not evaluating the CD-NOD method (Huang et al., 2017), as we do not have access to a MatLab license. That is, we compare against FCI (Spirtes et al., 2013), GES (Chickering, 2002), NOTEARS (Zheng et al., 2018), DirectLinGAM (Shimizu et al., 2011), plus a random baseline.
Following (Guo et al., 2024a, Sec. 6), we describe the DGP in detail. The CdF parameters $\mathbf{N} = [\psi, \theta]$ were randomly generated with distinct and independent elements in each environment. Samples within each environment have the noise variables $\mathbf{S}$ generated via Laplace distributions conditioned on the corresponding CdF parameters—i.e., the CdF parameter is the location (mean) of the Laplace distribution. We observe a bivariate vector $\mathbf{X} = [X_1, X_2] \in \mathbb{R}^2$ and aim to uncover the causal direction between $X_1$ and $X_2$. Let the superscript $(\cdot)^e$ denote variables contained in environment $e$.

---

[5]https://github.com/syguo96/Causal-de-Finetti. Our code is available at https://github.com/rpatrik96/IEM

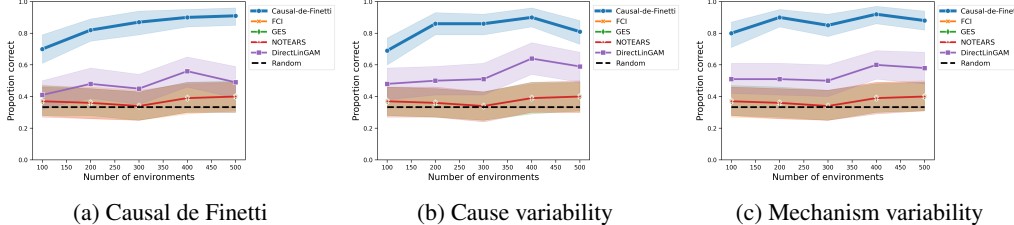

(a) Causal de Finetti      (b) Cause variability      (c) Mechanism variability

Figure 6: **Bivariate causal discovery is possible with cause and mechanism variability:** Comparison of the CdF protocol with FCI, GES, NOTEARS, DirectLinGAM, and a random baseline for causal structure discovery in the bivariate case with continuous random variables. The proportion of correctly identified causal structures is shown against a different number of environments, chosen from $\{100, 200, 300, 400, 500\}$. Shading shows the standard deviation across 100 seeds. **(a):** the original CdF setting, reproducing (Guo et al., 2024a, Fig. 3(a)) with non-delta priors for both CdF parameters; **(b):** cause variability with a delta parameter prior for the effect-given-the-cause parameter $\psi$; **(c):** mechanism variability with a delta parameter prior for the cause parameter $\theta$; For details, cf. Appx. F

Then, the data is generated as follows:

$$\mathbf{N}^e \sim \text{Uniform}[-1, 1] \tag{25}$$

$$\mathbf{S}^e \sim \text{Laplace}(\mathbf{N}, 1) \tag{26}$$

$$\mathbf{X}^e = \mathbf{A}^e \mathbf{S}^e + \mathbf{B}^e \left(\mathbf{N}^e\right)^{\circ 2} \mathbb{1}_{\text{nonlinear}}(e), \tag{27}$$

where $\circ 2$ denotes elementwise squaring. $\mathbf{A}^e \in \mathbb{R}^{2 \times 2}$ is a randomly sampled triangular matrix and $\mathbf{B}^e = \mathbf{A}^e - \mathbf{I}$, where $\mathbf{I}$ is the identity matrix. The causal direction is randomly sampled from $X_1 \to X_2, X_2 \to X_1, X_1 \perp X_2$—this ensures that $\mathbf{A}$ is either a lower triangular, upper triangular or diagonal matrix. $\mathbb{1}_{\text{nonlinear}}(e)$ is an environment-dependent, randomly sampled nonlinear-dependence indicator, which models the realistic scenario of invariant causal structure but changing functional relationships.

We implement cause and mechanism variability in the above synthetic DGP, which by changing the `scm_bivariate_continuous` function in the original GitHub repository. There, we set the noise variables for the delta-distributed CdF parameter ($\theta$ for mechanism and $\psi$ for cause variability) to be equal to the corresponding parameter value (as that is used as the location of the Laplace distribution). This means collapsing the Laplace distribution to a delta distribution for the corresponding CdF parameter in (26).

For comparison, we evaluate three settings: the original scenario (with non-delta priors for both parameters), cause variability, and mechanism variability. We use, as in the original code, two samples per environment and ablate over $\{100, 200, 300, 400, 500\}$ environments. Each experiment is repeated 100 times. We measure causal structure identification by three conditional independence tests with a significance level of $\alpha = 0.05$. We choose the estimated causal structure to be the one corresponding to the test with the highest $p$-value.

**Results.** Fig. 6 shows the proportion of correctly identified causal structures for different numbers of environments. The Causal-de-Finetti algorithm outperforms all the other methods with an accuracy close to $100\%$. This holds not just in the original scenario proposed by Guo et al. (2024a) (Fig. 6a), but also in the case of cause and mechanism variability (Figs. 6b and 6c), corroborating our Thm. 2.

## G ACRONYMS

**ANM** Additive Noise Model

**CD** Causal Discovery
**CdF** Causal de Finetti
**CRL** Causal Representation Learning

**DAG** Directed Acyclic Graph
**DGP** data generating process

**GCL** Generalized Contrastive Learning

**i.i.d.** independent and identically distributed
**ICA** Independent Component Analysis
**ICM** Independent Causal Mechanisms

**IEM** Identifiable Exchangeable Mechanisms

**LVM** latent variable model

**MSS** Mechanism Shift Score

**OOD** out-of-distribution

**RV** random variable

**SEM** Structural Equation Model
**SMS** Sparse Mechanism Shift

**TCL** Time-Contrastive Learning

