# OpenReview forum: "Identifiable Exchangeable Mechanisms for Causal Structure and Representation Learning"
_ICLR.cc/2025/Conference — ICLR 2025 Spotlight_

### Official Review · Reviewer_anYP · 2024-10-25

**Soundness:** 4
**Presentation:** 3
**Contribution:** 4
**Rating:** 6
**Confidence:** 3

**Summary:**

This paper introduces a novel framework called Identifiable Exchangeable Mechanisms (IEM). IEM aims to unify approaches in Causal Discovery (CD), Independent Component Analysis (ICA), and Causal Representation Learning (CRL) for identifying causal structures and learning representations. By extending the Causal de Finetti (CdF) theorem, IEM introduces the concepts of "cause variability" and "mechanism variability." This framework relaxes the assumption of independent and identically distributed (i.i.d.) data, allowing for causal structure and representation identification under broader conditions. Additionally, it establishes a new duality in ICA techniques, incorporating Time-Contrastive Learning (TCL) and other auxiliary variables.

**Strengths:**

The IEM framework, which extends the CdF theorem to provide a theoretical foundation for identifying causal structures based on exchangeable data, is a significant contribution bridging the fields of causal inference and representation learning. By unifying Causal Discovery (CD), Independent Component Analysis (ICA), and Causal Representation Learning (CRL) within a cohesive framework, it offers a straightforward and intuitively convincing approach. This paper is expected to play an essential role, particularly as causal representation learning continues to advance.

**Weaknesses:**

The paper attempts to present a wide range of theories within a limited space, which results in a somewhat abstract discussion. While readers well-versed in causal discovery might follow its value, those less familiar with the area may struggle to appreciate the paper’s contributions fully. Although it may be challenging, adding concrete examples to illustrate the unification of CD, ICA, and CRL would likely make the framework’s significance clearer and broaden its appeal.

Additionally, a few minor issues are noticeable:

- In line 107, "tequence's" appears to be a typo and should be "sequence's."
- In Equation (6), $p(y^i|^i)$ might be missing the term $\psi$.
- In the first sentence of Theorem 2, there is a misplaced bracket (]).

**Questions:**

Would it be possible to illustrate the significance of this paper with concrete examples?

---

> ### Author Response · Authors · 2024-11-18
>
> We thank Reviewer anYP for praising our **“significant contribution bridging the fields of causal inference and representation learning”**, our **“straightforward and intuitively convincing approach”**, and the **“expected essential role”** our contribution might play in CRL. We also thank you for pointing out the inconsistencies and typos, which we corrected in the new version.
>
> ### Concrete examples
> Thank you for pointing out the merit of concrete examples to provide intuition about IEM. We now provide intuition and **real-world examples in Appendix E** to demonstrate how the assumptions for the IEM framework, particularly for cause and mechanism variability, can model realistic scenarios.
> The **intuition** behind these conditions relies on the Sparse Mechanism Shift hypothesis from [Perry et al., 2022](http://arxiv.org/abs/2206.02013), which posits that only a subset of causal mechanisms changes between environments.
>
> Our examples show how only a subset of the causal mechanisms changes realistically across environments, including simple climate models, natural experiments in economics, and medical examples. For example, natural experiments in economics are based on selecting two sets of populations with (approximately) the same environmental and social factors, but with a difference in the relevant economic policies (e.g., a different local tax policy applies). This example belongs to mechanism variability, as the populations are the same.

---

> > ### Comment · Reviewer_anYP · 2024-11-18
> >
> > Thank you! These are very clear examples and will help a wide range of readers understand.

---

> > > ### Author Response · Authors · 2024-11-19
> > >
> > > Thank you so much for your feedback regarding the examples. Is there anything else we can improve to make our submission more worthy of acceptance?

---

### Official Review · Reviewer_Ruk1 · 2024-11-04

**Soundness:** 3
**Presentation:** 3
**Contribution:** 3
**Rating:** 8
**Confidence:** 4

**Summary:**

This work introduces a unified framework that formulates causal mechanisms and causes through a new class of identifiable exchangeable mechanisms, addressing the identifiability of latent noise variables, latent causal variables z, and the latent structure among z. This framework centers on two key sources of variability—cause variability and mechanism variability. By analyzing these variabilities, this work provides a rigorous approach to identifying both latent noise/causal variables and the latent structure among these causal variables.

**Strengths:**

1) This work addresses a challenging problem in causal representation learning, namely identifiability analysis.

2) The proposed framework is novel and provides a valuable perspective for examining the relationship between cause variability, mechanism variability, and the identifiability of latent noise, latent causal variables, and the associated causal structure.

3) TCL is a very elegant and simple framework for nonlinear ICA. The analysis connecting identifiable exchangeable mechanisms to temporal contrastive learning (TCL) is insightful. Notably, this work introduces a novel approach to identify latent noise variables by analyzing changes in the mixing mapping from latents to observed data, distinguishing it from traditional nonlinear ICA, which typically relies on changes in the distributions of latents.

4) Many insights from this work are likely to be highly significant for advancing our understanding of causal representation learning.

**Weaknesses:**

Overall, this work is highly interesting and provides valuable insights that advance our understanding of causal representation learning. The proposed framework is novel, and its discussion on identifiability offers a fresh perspective on key challenges in the field. I appreciate the significant insights presented, which have the potential to drive progress in this area. Below, I outline some concerns to discuss with the authors:

1) In general, for causal discovery in multi-domain data, the causal structure can be identified using the principle of Independent Causal Mechanisms (ICM). Specifically, in the causal direction, the parameters of the causal mechanism tend to be independent, whereas in the reverse direction, these parameters show dependency. Thus, ICM can be used to determine causal direction (see [1,2]). Furthermore, if latent causal variables can be identified by leveraging both cause variability and mechanism variability, then the causal structure should also become evident. Given this, is it necessary to discuss causal structure separately once latent causal variables have been identified? This also suggests a possible connection between the conditions required to identify latent causal variables and those needed to identify causal structure.

2) I find the concept of the duality between cause and mechanism variability in TCL fascinating. It's particularly interesting to consider changes in the mixing mapping rather than changes in the distribution of latent variables. Are there any insights into real-world applications for this setting?

3) Equations (11) for CauCA and (5) for Liu et al., 2024 represent two distinct yet crucial conditions for identifiability, both relying on sufficient changes along the edges. Are there any new insights from the framework of identifiable exchangeable mechanisms that could shed light on these different formulations of edge changes? (Note that CauCA does not guarantee identifiability of the exogenous variables.)



a. In line 78, 'GCL' is introduced for the first time; please expand the acronym upon its first mention.

b. In line 95, two different mappings, specifically from s to o and from z to o, are defined using the same function f. It may be clearer to use distinct notation for these mappings.

c. Please ensure consistency in the notation for \psi in Equation (3) and in line 131.

d. \theat^{g}_{i0} has not been defined.

e. In Equation (6), the term p(y∣???) needs clarification."

[1] Ghassami, AmirEmad, et al. "Multi-domain causal structure learning in linear systems." Advances in neural information processing systems 31 (2018).

[2] Huang, Biwei, et al. "Causal discovery from heterogeneous/nonstationary data." Journal of Machine Learning Research 21.89 (2020): 1-53.

I am happy to raise my rating once the above concerns have been addressed

**Questions:**

See above

---

> ### Author Response · Authors · 2024-11-18
>
> We thank Reviewer Ruk1 for praising **our contribution's elegance and its valuable perspective, and deeming its impact highly significant!** We also appreciate the constructive and detailed feedback, which we will address below. We also addressed the notational issues (a-e) in our new submission.
>
> ### Causal structure and causal variables
> We thank the Reviewer’s insightful question about the relation between identifying causal structure and causal variables.
> As [Wendong et al., 2023](http://arxiv.org/abs/2305.17225) noted in their Causal Component Analysis papers, the two notions are related, and required for causal representation learning. However, **identifying the causal variables does not necessarily imply the identification of the causal structure**.
> One might see the motivation for CauCA as admitting this nontrivial relationship - as an *example* consider that even if we learned all causal variables X, Y, Z correctly, this does not imply a single causal structure, not even the topological ordering can be determined from knowing only X, Y, and Z.
> However, by leveraging, e.g., the learned functional relationships, one could potentially apply the partial derivative-based approach from [Reizinger et al., 2023](https://openreview.net/forum?id=2Yo9xqR6Ab) to extract the causal structure. In short, knowing the causal variables and the causal structure is related, though not equivalent.
>
> ### Real-world insights from the duality of cause and mechanism variability for TCL
> At the end of Section 3.3, we provide a medical example to illustrate under what circumstances the duality could prove useful - please also refer to the **real-world examples** of cause and mechanism variability we provide in **Appendix E**.
> Thus, we think that our **proposed duality could open up new identifiability results**, yielding to novel representation learning methods to so far not addressed realistic scenarios.
>
> ### Insights for the edge-based conditions of CauCA and [Liu et al., 2024](http://arxiv.org/abs/2403.15711)
> We thank for the insightful question! We **compare the two assumptions in Appendix A.7**, concluding that the condition of [Liu et al., 2024](http://arxiv.org/abs/2403.15711) can be seen as *“emulating” perfect interventions*. That is, it is stronger than the assumption of CauCA. As the Reviewer noted, the identifiability guarantee of Liu et al., 2024 is also stronger as it additionally identifies the exogenous variables. Thus, it is unsurprising that they require a stronger assumption.
> The additional high-level insight IEM provides for these assumptions is that edge-based conditions can be approached from a passive (distribution shift) and an active (interventional) perspective.

---

> > ### Comment · Reviewer_Ruk1 · 2024-11-18
> > **Thank you for the response.**
> >
> > My concerns have been addressed, and I have raised my rating to 8.

---

> > > ### Author Response · Authors · 2024-11-18
> > >
> > > Thank you so much!

---

### Official Review · Reviewer_B5R6 · 2024-11-08

**Soundness:** 3
**Presentation:** 3
**Contribution:** 3
**Rating:** 8
**Confidence:** 3

**Summary:**

The paper “Identifiable Exchangeable Mechanisms for Causal Structure and Representation Learning” proposes the Identifiable Exchangeable Mechanisms (IEM) framework to unify causal discovery (CD), Independent Component Analysis (ICA), and Causal Representation Learning (CRL) in the context of exchangeable data. This framework generalizes the Causal de Finetti theorem (Guo et al., 2022) by introducing relaxed identifiability conditions called cause and mechanism variability. These new conditions aim to enable causal structure identification across non-i.i.d. datasets, thus extending traditional causal inference to handle multi-environment data. The authors theoretically derive and discuss these conditions and provide conceptual examples to illustrate the IEM framework's utility.

**Strengths:**

The IEM framework proposed in this paper provides a novel theoretical contribution by unifying causal discovery, ICA, and CRL using exchangeability and probabilistic graphical models. The introduction of cause and mechanism variability as identifiability conditions broadens the applicability of causal inference to non-i.i.d. data. This relaxation of traditional i.i.d. assumptions aligns well with current trends in causality research, which increasingly focus on real-world, structured data that deviate from idealized assumptions, thus contributing to the field's practical impact.

**Weaknesses:**

1.	The paper lacks empirical validation on either synthetic or real-world datasets. Given that the IEM framework is proposed as a practical tool for identifying causal structures in multi-environment data, empirical tests on structured datasets would significantly bolster its practical relevance and demonstrate its utility across different applications.

2.	While theoretically motivated, the cause and mechanism variability assumptions remain abstract, and their practical applicability may be challenging for readers to conceptualize. Including concrete, real-world examples to illustrate where these assumptions hold would enhance the understanding of the framework's assumptions in practical scenarios.

3.	Although the IEM framework aims to unify CD, ICA, and CRL, the paper lacks a direct comparative analysis to benchmark IEM against established methods within these areas. A comparative discussion, even theoretical, could better highlight IEM’s unique contributions and clarify where it stands relative to traditional ICA, causal discovery, or representation learning methods.

**Questions:**

1.	I suggest that the authors provide specific, real-world scenarios where the assumptions of cause and mechanism variability would hold, to help contextualize these conditions.

2.	Has the framework been considered for use on any real-world datasets, such as those in biology or social sciences, to evaluate its generalization and robustness in non-idealized data?

3.	Are there foreseeable extensions of IEM to structured but non-exchangeable data (such as time series), which would broaden its applicability further?

---

> ### Author Response · Authors · 2024-11-18
>
> We thank Reviewer B5R6 for praising **our contribution’s novelty and our contribution to relax idealized assumptions in causality research**. We also thank you for the detailed remarks and constructive feedback. We hope that our answers below can sufficiently address your concerns.
>
> ### Empirical evaluation
> Based on the reviewer’s feedback, we implemented a **synthetic evaluation protocol**, based on the publicly available repository of the Causal de Finetti paper (to be found at https://github.com/syguo96/Causal-de-Finetti). We focus on the continuous case, as our proofs hold for that scenario, and implement cause and mechanism variability. The continuous experiments used in the original CdF paper consider the bivariate case, which we follow to be comparable. The only change in the evaluation protocol is not evaluating the CD-NOD method, as we do not have access to a MatLab license.
>
> We include the results of causal structure identification in our revised manuscript in Appendix F in Fig. 6. Our **evaluation corroborates our Thm. 2**, demonstrating that if the data is generated with a **cause (Fig. 6b) or mechanism variability** assumption (Fig. 6c), then **causal structure identification is possible**. Furthermore, the accuracy of the Causal de Finetti method does not significantly diminish compared to the original setting (Fig. 6a) and is still **substantially better than the other methods** (NOTEARS, DirectLinGAM, FCI, GES), showing that cause or mechanism variability is sufficient for causal discovery.
>
> ### Real-world examples for cause and mechanism variability and intuition
> Thank you for pointing out the merit of concrete examples to provide intuition about IEM. Beyond the synthetic experiments, we also provide **real-world examples in Appendix E** to demonstrate how the assumptions for the IEM framework, particularly cause and mechanism variability, can model realistic scenarios. Our examples include ones from *economics, healthcare, and simple environmental models*.
> We also discuss the intuition behind cause and mechanism variability. which relies on the Sparse Mechanism Shift hypothesis [(Perry et al., 2022)](http://arxiv.org/abs/2206.02013).
>
> Our examples show how only a subset of the causal mechanisms changes realistically across environments, including the simple climate models, natural experiments in economics, and medical examples.
>
> ### Applications to real-world data sets
> As our real-world examples from the previous paragraph show, we believe that the IEM framework can model realistic medical and socioeconomic data. Thus, we desire to transfer our insights to applications for which we would first need access to such data sets.
>
> ### Extensions to structured but non-exchangeable data
> That is a great question! As **IEM** encompasses TCL, it **can already model a specific type of temporal data**. Based on this, we would conjecture that further extensions to other data structures is possible.
>
> ### Direct comparative analysis
> In our new experiments, we compare IEM to other methods from the causality literature.
> Theoretically, we would like to clarify that the goal of proposing IEM was to provide a unifying framework. That is, many causal discovery, traditional ICA, and causal representation learning methods are encompassed by IEM; IEM is not a particular method we can compare to. Where IEM provides new results - for example, with the relaxation of the Causal de Finetti theorem -, we provided an experimental comparison (please refer to Appendix F).
> We kindly ask the reviewer to clarify their point if we misunderstood it.

---

> ### Author Response · Authors · 2024-11-25
>
> Dear Reviewer B5R6,
>
> Thank you again for your constructive feedback! We have updated our submission with experimental evaluation, and described real-world examples to illustrate our framework's assumptions (particularly regarding cause and mechanism variability). Please let us know if you have any further questions.

---

### Official Review · Reviewer_pAxk · 2024-11-08

**Soundness:** 2
**Presentation:** 1
**Contribution:** 2
**Rating:** 6
**Confidence:** 2

**Summary:**

From the perspective of IEM, the paper uses the identifiable exchangeable mechanisms framework that unifies causal discovery, independent component analysis and causal representation learning. The authors further shows how exchangeability can model out-of-distribution and interventional distribution.

**Strengths:**

The paper provides an analysis of causal discovery, independent component analysis and causal representation learning through the framework of IEM, Providing theoretical connections among these methods. This provides researchers deeper understanding of the assumptions and inter-connections among these methods.

**Weaknesses:**

1. The arguments in section 3 seem to be loosely connected, and not enough time was spent tying these sections together. I would expect more detail in explaining how this framework can benefit the development of these fields. Section 4 briefly discusses how the framework could be used to unify research in CD, ICA, and CRL but then shifts to discussing OOD and interventional distributions. A more in-depth discussion on the connections among these fields and how advancements in one field could benefit others would be beneficial.

2. Despite the strong theoretical elements in the paper, it lacks specific examples. In the case studies for each subsection, it would be better if the authors could provide examples or counterexamples, along with a more detailed discussion of these examples/test cases, so that readers can gain a clearer understanding.

**Questions:**

Could you elaborate more on the framework of IEM and the exchangeability of data? Specifically, could you explain, with examples, the physical intuition behind exchangeable sequences? Moreover, the variable $Z$ is referred to multiple times as a causal variable. Is there a rigorous definition of this variable, and how can a variable contain cause-effect relationships?

---

> ### Author Response · Authors · 2024-11-18
>
> We thank Reviewer pAxk for providing feedback on our manuscript and praising our submission’s **contribution towards a deeper understanding of how causal discovery, Independent Component Analysis, and causal representation learning methods relate to each other**. In the following, we address the Reviewer’s concerns.
>
> ### Sections 3 & 4
> Reviewers szvU, anYP, and Ruk1 praised our clear writing and the paper’s clear structure: **“The paper is well-structured and clearly written. I appreciate a broad Discussion and future directions section.”** (Reviewer szVU) **“straightforward and intuitively convincing approach”** (Reviewer anYP), **“TCL is a very elegant and simple framework for nonlinear ICA. The analysis connecting identifiable exchangeable mechanisms to temporal contrastive learning (TCL) is insightful.”** (Reviewer Ruk1).
> These statements respectfully disagree with the Reviewer’s criticism of the parts Section 3 being loosely connected. We devoted significant time to providing a unified structure not just to our theoretical framework but also to our writing. Notably, each subsection starts with a probabilistic model, then provides a case study by describing a well-known algorithm in that field, and ends with our new, more general results.
>
> The **first paragraph of Section 4 starts with explicitly answering** the Reviewer’s question about **what do the fields of causal discovery, ICA, and causal representation learning gain from our IEM framework**.
> - We mention the active-passive view (i.e., whether data is collected or considered as given);
> - We end Section 4 with a discussion of how IEM can enable identifying components of causal systems, which might lead to more data-efficient algorithms (Sec 4.3);
> - We also pinpoint potential gaps, uncovered by the duality of cause and mechanism variability (Sec 4.4);
> - Furthermore, the last paragraph in Section 3 also shows how the field can benefit from IEM.
>
> Section 4 indeed discusses OOD generalization and interventional distributions, as those are the instantiations which constitute exchangeability in causal discovery, ICA, and causal representation learning.
>
> ### Specific examples
> Thank you for pointing out the merit of concrete examples to provide intuition about IEM. We added **intuition and real-world examples in Appendix E** to demonstrate how the assumptions for the IEM framework, particularly cause and mechanism variability, can model realistic scenarios.
>
> ### Causal variables
>
> > Moreover, the variable Z  is referred to multiple times as a causal variable. Is there a rigorous definition of this variable, and how can a variable contain cause-effect relationships?
>
> Causal variables are defined as the right-hand side of structural equations in Structural Equation Models (SEMs), for example, as described in Eq (12) in [(Schölkopf et al., 2021)](http://arxiv.org/abs/2102.11107). The causal variable can only contain the topological order, which, by convention [(Pearl, 2009)](http://ebooks.cambridge.org/ref/id/CBO9780511803161), means that for the vector variable $
> \mathbf{Z}$ for each $i<j$, only $Z_i$ can be a parent of $Z_j$ - i.e., only $Z_i$ can be present on the left-hand side in the structural equation expressing $Z_j$.
> That is, the causal variables on their own are not enough to provide all causal information. To express the cause-effect relationships, a DAG is used, whereas the functional relationships are described by the SEM equations (the partial derivatives of the equations also give back the DAG as for non-existing edges, the partial derivatives will be zero).

---

> > ### Author Response · Authors · 2024-11-18
> >
> > ### Elaboration on IEM and exchangeability
> > > Could you elaborate more on the framework of IEM and the exchangeability of data?
> >
> > We thank the reviewer for the question and would like to ask them to follow up in case our elaboration takes a different direction as expected by the reviewer, and we ask for a more specific formulation.
> >
> > **Exchangeability is a statistical notion that can model specific data**, such as data coming from multiple environments - which is prevalent in the causality and identifiable representation learning literatures. Intuitively, we can describe exchangeable data as a **mixture of i.i.d. components, where data for each component is drawn from a different distribution with potentially different (de Finetti) parameters**. We would like to highlight that *the conventional definition of exchangeability (formulated by permutation invariance) is not intuitive*. This is why we opted for the “mixture of i.i.d.” approach, as that relates ore to the modeling choices in machine learning.
> >
> > **The IEM framework is our contribution to show how exchangeability is the underlying principle of many identifiability results in causal discovery,Independent Component Analysis, and causal  representation learning.**
> >
> > > Specifically, could you explain, with examples, the physical intuition behind exchangeable sequences?
> >
> > The simplest examples of exchangeable models are **urn models**: A random number of blue and red balls are placed in an urn, which is hidden from the observer, we don’t know how many blue and red balls it contains. Balls are then drawn with replacement from the urn, and presented to the observer. The sequence of colours of the subsequent draws constitute the exchangeable process. These random variables are not i.i.d. since the first balls drawn provide information about the composition of balls in the urn, and thus allows you to predict the colours you will see in the future. The more blue balls you see among your first observations, the more probable it is - conditioned on these observations - that the next ball is also blue.
> >
> > **Unlike i.i.d. models, exchangeable statistical models naturally capture the fact that past observations provide information about future observations**. The exchangeability requirement (invariance to permutations) ensures that the information the past provides bout the future is only ‘correlational’ and not ‘causal’: observing a blue ball does not cause the next ball to be blue, it merely informs about the likelihood of it being blue.
> > *Since in these models subsequent observations are no longer statistically independent, exchangeable probabilstic models have more intricate conditional independence structure*: CI relationships that trivially hold in all i.i.d. process hold in some, but not all exchangeable processes. This richer independence structure in turn provides a clearer reflection of the underlying causal relationships, thus enabling the strong identifiability results we claim.
> >
> > **A more realistic example** is the one [Guo et al., 2024](https://arxiv.org/abs/2203.15756) use in their Causal de Finetti paper (it is from page 5, below their Theorem 2). We paraphrase their example. Assume a *hospital with are two patients*. A patient’s symptom is the cause of a doctor’s diagnosis. Suppose we are interested in predicting a patient’s diagnosis given their symptoms. We model this scenario as a causal graph akin to the one in our Fig. 2a, where $X$ stands for the patients’ symptoms, $Y$ for the diagnoses, $\theta$ for the hospital’s standard of care, $\psi$ for how one doctor diagnoses a patient. The conditional independence formulated by exchangeability says knowing another patient’s symptom will not help us to predict the diagnosis of this patient if we know this patient’s symptoms already. The conditional independence thus formulated the intuition behind causal and anti-causal problem in the language of probability: the distribution of the cause, other patients’ symptoms in this case, will not help
> > prediction about the effect given cause, i.e., one patient’s diagnosis given his own symptoms.
> >
> > **We provide additional examples for cause and mechanism variability in Appendix E.**

---

> > > ### Comment · Reviewer_pAxk · 2024-11-24
> > > **Thank you for the response and additional examples and details**
> > >
> > > I understand the work better with more details and examples. I have raised my score.

---

### Official Review · Reviewer_vkuV · 2024-11-09

**Soundness:** 3
**Presentation:** 2
**Contribution:** 2
**Rating:** 5
**Confidence:** 3

**Summary:**

This paper introduces the Identifiable Exchangeable Mechanisms (IEM) framework, which unifies key methods in causal structure discovery, Independent Component Analysis (ICA), and causal representation learning within a single probabilistic graphical model. The authors use the IEM framework to generalize the Causal de Finetti theorem, relaxing the necessary conditions for causal structure identification in exchangeable data. They introduce new conditions termed "cause and mechanism variability" and provide novel identification results.

**Strengths:**

1. The paper effectively bridges the relatively different identifiable methods of causal structure identification and representation learning by introducing the IEM framework. This framework offers a novel, theoretically grounded perspective that could be influential for identifying latent structures and causal mechanisms.
2. By generalizing the Causal de Finetti theorem, the authors relax traditional assumptions and derive new identification conditions for causal structures.

**Weaknesses:**

Although the theoretical contribution is rigorous, some experiments are still needed to evaluate the effectiveness of the proposed IEM framework.

**Questions:**

See above.

---

> ### Author Response · Authors · 2024-11-18
>
> We thank Reviewer vkuV for praising our **“novel, theoretically grounded perspective that could be influential for identifying latent structures and causal mechanisms”** and our **new identification results**.
>
> ### Experimental evaluation
> Based on the reviewer’s feedback, we implemented a **synthetic evaluation protocol**, based on the publicly available repository of the Causal de Finetti paper (to be found at https://github.com/syguo96/Causal-de-Finetti). We focus on the continuous case, as our proofs hold for that scenario, and implement cause and mechanism variability. The continuous experiments used in the original CdF paper consider the bivariate case, which we follow to be comparable. The only change in the evaluation protocol is not evaluating the CD-NOD method, as we do not have access to a MatLab license.
>
> We include the results of causal structure identification in our revised manuscript in Appendix F in Fig. 6. Our **evaluation corroborates our Thm. 2**, demonstrating that if the data is generated with a **cause (Fig. 6b) or mechanism variability** assumption (Fig. 6c), then **causal structure identification is possible**. Furthermore, the accuracy of the Causal de Finetti method does not significantly diminish compared to the original setting (Fig. 6a) and is still **substantially better than the other methods** (NOTEARS, DirectLinGAM, FCI, GES), showing that cause or mechanism variability is sufficient for causal discovery.

---

> ### Author Response · Authors · 2024-11-25
>
> Dear Reviewer vkuV,
>
> Thank you again for your constructive feedback! We have updated our submission with experimental evaluation to illustrate our framework's practical applicability. Please let us know if you have any further questions.

---

> > ### Author Response · Authors · 2024-11-29
> >
> > Dear Reviewer vkuV,
> >
> > The author-reviewer discussion period ends soon. Please let us know if you require further clarification. We hope that based on our response and providing experimental results, you consider increasing your score.
> >
> > The authors

---

### Official Review · Reviewer_szvU · 2024-11-12

**Soundness:** 3
**Presentation:** 3
**Contribution:** 2
**Rating:** 6
**Confidence:** 2

**Summary:**

The paper introduces Identifiable Exchangeable Mechanisms (IEM), a general framework that unifies causal discovery (CD), independent component analysis (ICA), and causal representation learning (CRL). The IEM model relies on the exchangeable data (e.g., data from multiple environments), and generalizes the Causal de Finetti theorem. The authors showed, that some assumptions required by CD, ICA, and CRL are special cases of the proposed IEM model.

**Strengths:**

The introduced framework provides a unifying perspective on several multi-environment causal inference tasks: CD, ICA, and CRL.  The paper is well-structured and clearly written. I appreciate a broad Discussion and future directions section.

**Weaknesses:**

In my opinion, the major weakness of the paper is the lack of empirical evaluation or case studies for each of the mentioned settings (Sec 3.2, 3.3, 3.4). I encourage the authors to provide at least one synthetic/real-world dataset, where the IEM is applied in all three settings. I understand, that the main contribution of the paper is theoretic, but a case study can provide some applied insights, e.g., how degrees of non-i.i.d. in data facilitate a causal inference task at hand.

I am happy to raise my score, if this weakness is addressed during the rebuttal.

**Questions:**

- Does the proposed framework generalize to a setting where we allow for hidden confounds between causal variables?
- There seems to be an issue with the notation in Eq. 6?

---

> ### Author Response · Authors · 2024-11-18
>
> We thank Reviewer szvU for acknowledging our submission's **clear writing, good structure, and the extended Discussion**. We address the reviewer’s questions and criticisms in the following.
>
> ### Empirical evaluation
> Based on the reviewer’s feedback, we implemented a **synthetic evaluation protocol**, based on the publicly available repository of the Causal de Finetti paper (to be found at https://github.com/syguo96/Causal-de-Finetti). We focus on the continuous case, as our proofs hold for that scenario, and implement cause and mechanism variability. The continuous experiments used in the original CdF paper consider the bivariate case, which we follow to be comparable. The only change in the evaluation protocol is not evaluating the CD-NOD method, as we do not have access to a MatLab license.
>
> We include the results of causal structure identification in our revised manuscript in Appendix F in Fig. 6. Our **evaluation corroborates our Thm. 2**, demonstrating that if the data is generated with a **cause (Fig. 6b) or mechanism variability** assumption (Fig. 6c), then **causal structure identification is possible**. Furthermore, the accuracy of the Causal de Finetti method does not significantly diminish compared to the original setting (Fig. 6a) and is still **substantially better than the other methods** (NOTEARS, DirectLinGAM, FCI, GES), showing that cause or mechanism variability is sufficient for causal discovery.
>
> ### Insights from real-world examples
> Beyond the synthetic experiments, we also provide **real-world examples in Appendix E** to demonstrate how the assumptions for the IEM framework, particularly cause and mechanism variability, can model realistic scenarios. Our examples include ones from *economics, healthcare, and simple environmental models*.
>
> ### IEM for hidden confounders
> > Does the proposed framework generalize to a setting where we allow for hidden confounds between causal variables?
>
> This is a great question! In some settings, the answer is yes. Namely, the work by [Jin & Syrgkanis, 2023](http://arxiv.org/abs/2311.12267) proves three theoretical results (two for different linear SEMs, and one for nonparametric SEMs) without assuming faithfulness, i.e., latent confounders can be present.
>
>
> ### Eq 6.
> Thank you for pointing out the discrepancy, the equation was indeed missing the conditioning variables from the Markov factorization, the correct first term is $p(y^i|x^i, \psi)$. We corrected this in the new version of our submission.

---

> ### Author Response · Authors · 2024-11-25
>
> Dear Reviewer szvU,
>
> Thank you again for your constructive feedback! We have updated our submission with experimental evaluation and described real-world examples to illustrate our framework's assumptions and applicability. Please let us know if you have any further questions.

---

> > ### Comment · Reviewer_szvU · 2024-11-26
> >
> > Dear authors,
> >
> > Thank you for your response! I appreciate newly added experiments and real-world examples. Indeed, the proposed framework, IEM, is highly effective for the task of causal discovery in multiple environments. Nevertheless, the experimental evidence on **independent component analysis**  and **causal representation learning** is missing. Therefore, in my opinion, the paper is still incomplete, and I am retaining my current score.

---

### Author Response · Authors · 2024-11-18
**Joint response**

We thank all Reviewers for their time, constructive feedback and thoughtful questions. We are grateful that the Reviewers deemed our submission. We provide a summary of the questions and concerns that were mentioned by multiple Reviewers in our joint response here.

We are grateful that **Reviewer szvU** acknowledged our submission's *clear writing, good structure, and the extended Discussion*;  **Reviewer vkuV** praised our *“novel, theoretically grounded perspective that could be influential for identifying latent structures and causal mechanisms”* and our new identification results; **Reviewer pAxk** praised our submission’s *contribution towards a deeper understanding of how causal discovery, Independent Component Analysis, and causal representation learning methods relate to each other*; **Reviewer B5R6** highlighted our contribution’s *novelty and our contribution to relax idealized assumptions in causality research*; **Reviewer Ruk1** praised our contribution's *elegance and its valuable perspective, and deeming its impact highly significant*; and **Reviewer anYP** emphasized our *“significant contribution bridging the fields of causal inference and representation learning”*, our *“straightforward and intuitively convincing approach”*, and the *“expected essential role”* our contribution might play in CRL.

### Changes in the submission
We added to Appendices to elaborate the Reviewers’ requests:
- **Appendix E** details the **intuition behind cause and mechanism variability** by relating it to the Sparse Mechanism Shift hypothesis of [Perry et al., 2022](http://arxiv.org/abs/2206.02013) and provides **real-world examples** for both cases.
- **Appendix F** show **experimentally**, in the synthetic case, that cause and mechanism variability can lead to causal structure identification (please also refer to our Figure 6 in Appendix F)

### Case studies, real-world examples, and intuition
Thank you for pointing out the merit of concrete examples to provide intuition about IEM. We now provide intuition and **real-world examples in Appendix E** to demonstrate how the assumptions for the IEM framework, particularly for cause and mechanism variability, can model realistic scenarios.
The **intuition** behind these conditions relies on the Sparse Mechanism Shift hypothesis from [Perry et al., 2022](http://arxiv.org/abs/2206.02013), which posits that only a subset of causal mechanisms changes between environments.

Our examples show how only a subset of the causal mechanisms changes realistically across environments, including simple climate models, natural experiments in economics, and medical examples. For example, natural experiments in economics are based on selecting two sets of populations with (approximately) the same environmental and social factors, but with a difference in the relevant economic policies (e.g., a different local tax policy applies). This example belongs to mechanism variability, as the populations are the same.

### Empirical evaluation
Based on the Reviewers’ feedback, we implemented a **synthetic evaluation protocol**, based on the publicly available repository of the Causal de Finetti paper (to be found at https://github.com/syguo96/Causal-de-Finetti, we **added the code we used as supplementary material**). We focus on the continuous case, as our proofs hold for that scenario, and implement cause and mechanism variability. The continuous experiments used in the original CdF paper consider the bivariate case, which we follow to be comparable. The only change in the evaluation protocol is not evaluating the CD-NOD method, as we do not have access to a MatLab license.

We include the results of causal structure identification in our revised manuscript in Appendix F in Fig. 6. Our **evaluation corroborates our Thm. 2**, demonstrating that if the data is generated with a **cause (Fig. 6b) or mechanism variability** assumption (Fig. 6c), then **causal structure identification is possible**. Furthermore, the accuracy of the Causal de Finetti method does not significantly diminish compared to the original setting (Fig. 6a) and is still **substantially better than the other methods** (NOTEARS, DirectLinGAM, FCI, GES), showing that cause or mechanism variability is sufficient for causal discovery.

---

### Meta-Review · Area_Chair_sHR6 · 2024-12-20

**Metareview:**

This work propose and analyse a novel framework, Identifiable Exchangeable Mechanisms (IEM), to unify a set of important and prevalent schemes including causal discovery (CD), Independent Component Analysis (ICA), and Causal Representation Learning (CRL).
As the reviewers reach the consensus that the contribution of proposing the framework is novel and substantial, unifying the sets of identifiable methods provides important insights understanding the connections among these methods, and the paper is well written, it would be beneficial for our community to see such contribution and highlight at the ICLR conference.

**Additional Comments On Reviewer Discussion:**

The discussion resolves most of reviewers' concerns and provides additional insights and clarifications.

---

### Decision · Program_Chairs · 2025-01-22

Accept (Spotlight)